# DELVING INTO FEATURE SPACE: IMPROVING ADVERSARIAL ROBUSTNESS BY FEATURE SPECTRAL REGULARIZATION

## ABSTRACT

The study of adversarial examples in deep neural networks has attracted great attention. Numerous methods are proposed to eliminate the gap of features between natural examples and adversarial examples. Nevertheless, every feature may play a different role in adversarial robustness. It is worth exploring which feature is more beneficial for robustness. In this paper, we delve into this problem from the perspective of spectral analysis in feature space. We define a new metric to measure the change of features along eigenvectors under adversarial attacks. One key finding is that eigenvectors with smaller eigenvalues are more non-robust, i.e., adversary adds more components along such directions. We attribute this phenomenon to the dominance of the top eigenvalues. To alleviate this problem, we propose a method called *Feature Spectral Regularization (FSR)* to penalize the largest eigenvalue, and as a result, the other smaller eigenvalues get increased relatively. Comprehensive experiments demonstrate that FSR is effective to alleviate the dominance of larger eigenvalues and improve adversarial robustness on different datasets. Our codes will be publicly available soon.

## 1 INTRODUCTION

It is shown that the performance of Deep Neural Networks (DNNs) decreases dramatically when confronted with adversarial examples (Biggio et al., 2013; Szegedy et al., 2013; Goodfellow et al., 2015). The vulnerability of DNNs brings potential risk to safety-critical deployments (Finlayson et al., 2019). To mitigate this vulnerability of DNNs, numerous methods are proposed to improve adversarial robustness (Papernot et al., 2016b; Madry et al., 2018; Xie et al., 2019). Among these, adversarial training (AT) (Madry et al., 2018) is the most effective approach that achieves state-of-the-art performance under various attacks (Croce et al., 2020). Different from standard training, adversarial training trains DNNs on adversarial examples rather than natural examples.

There are numerous methods proposed to eliminate the gap of features between natural examples and adversarial examples (Kannan et al., 2018; Zhang & Wang, 2019; Zhang et al., 2019). However, they constrain the features on the whole without considering the distinction of contribution from an individual feature. This may be inappropriate since every feature may play a different role in robustness. The work of (Ilyas et al., 2019) argued that adversarial examples result from non-robust features (Ilyas et al., 2019), i.e., well-generalizing but brittle features. This inspires us that we could treat each feature separately in the adversarial setting. While the work of (Bai et al., 2021; Yan et al., 2021) considered the influence of different channels on robustness, the problem has not been explored clearly from the perspective of spectral signatures, i.e., the eigenvalues and eigenvectors of feature covariance. The space is split into many spectral components, and then it is worth analyzing which component is beneficial for adversarial robustness and which is fragile under attack.

In this paper, we show that spectral signatures of deep features have a close connection with adversarial robustness. Through applying principal component analysis (PCA) to deep features, we could split feature space to various eigenvectors with its eigenvalues. Our motivation comes from the phenomenon that the standard trained model often results in a sharp distribution of eigenvalues (Yu et al., 2020), i.e., the eigenvalues rapidly become very small as shown in Figure 1. This property may be beneficial for natural accuracy (Lezama et al., 2018; Papyan et al., 2020), while its

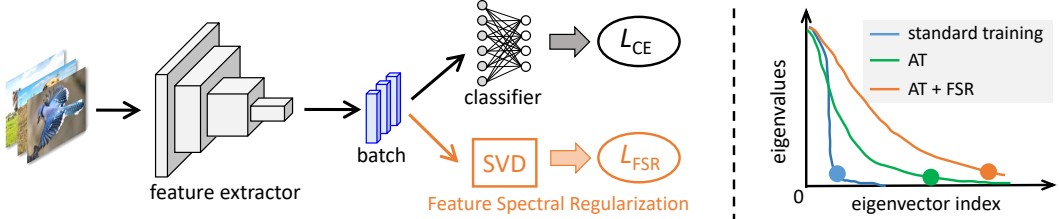

Figure 1: The architecture of DNN with Feature Spectral Regularization (FSR). FSR alleviates the dominance of top eigenvalues and enhances the role of relatively smaller eigenvalues in classification. When combined with AT, FSR further help learn more diverse features with higher dimensions. In the curve, the solid circles denote the estimated intrinsic dimension of features.

impact on robust generalization is unclear. However, we hypothesize that the sharp distribution of eigenvalues impels models to learn less diverse features and is a cause of the vulnerability of DNNs. A minority of eigenvalues occupy the overwhelming majority in the sum of eigenvalues, which may make the model pay little attention to features along the eigenvectors with smaller eigenvalues, so features along such directions are not generalized during training. To verify our hypothesis, we define a new metric to measure the variation of features along different eigenvectors under attacks, as shown in Figure 4. Our observation reveals that the adversary tends to add more components along eigenvectors with smaller eigenvalues, and such huge variation could be distinctly alleviated by AT.

Therefore, we propose to improve adversarial robustness by alleviating the sharp distribution of eigenvalues. Considering that more eigenvalues should occupy the majority, we propose a regularizer named Feature Spectral Regularization (FSR) to penalize the largest eigenvalue of the feature matrix covariance. Empirical evidence shows that FSR increases the overall eigenvalues relatively, making models focus on more spectral components during training. We also provide a theoretical explanation with robust linear regression. Comprehensive experiments confirm that FSR indeed improves the adversarial robustness on several datasets. Our contributions are summarized as follows:

- We find a close connection between spectral signatures of features and adversarial robustness. On one hand, standard trained model presents a sharp distribution of eigenvalues, which is beneficial for natural accuracy while harmful in adversarial setting. On the other hand, the adversary tends to add more quantity along eigenvectors with smaller eigenvalues.

- We propose Feature Spectral Regularization (FSR) to increase the overall eigenvalues relatively in deep feature space, thus alleviating the sharp distribution of eigenvalues. Furthermore, we provide a theoretical explanation based on robust linear regression.

- We empirically show that FSR improves adversarial robustness and alleviates the sharp distribution of eigenvalues, through comprehensive experiments.

## 2 RELATED WORK

**Adversarial Defense.** Many defense methods have been proposed to improve adversarial robustness since the discovery of adversarial examples (Papernot et al., 2016b; Xie et al., 2019; Carmon et al., 2019; Zhang et al., 2020). However, many of them are proven to be noneffective because they highly depend on obfuscated gradients (Athalye et al., 2018) or gradient masking (Papernot et al., 2017). Among these, adversarial training (Madry et al., 2018) is now regarded as the state-of-the-art method (Rice et al., 2020; Pang et al., 2021). Distinguished form standard training, adversarial training trains DNN on adversarial examples:

$$\min_{\theta} \max_{\|\delta\| \leq \epsilon} \mathbb{E}_{(x,y) \in D} \ L_{CE}\left(x + \delta, y; \theta\right) \tag{1}$$

where, $D$ is the dataset, the parameters of DNN are denoted as $\theta$, $\delta$ means the perturbation within the $\epsilon$-ball, and $L_{CE}$ is the cross-entropy (CE) loss.

By introducing a trade-off between robustness and generalization, TRADES (Zhang et al., 2019) is another framework that reaches comparative robustness with AT. Among the proposed methods based on AT, Adversarial Weight Perturbation (AWP) (Wu et al., 2020) explicitly regularizes the flatness of weight loss landscape, and forms a double-perturbation mechanism, which shows huge improvement on adversarial robustness and alleviates the overfitting in AT (Rice et al., 2020).

**Spectral Signatures of Feature Representations.** Some studies have revealed that the spectral signatures of features influence the performance in various learning tasks. The spectral properties are crucial to detect backdoors (Tran et al., 2018; Hayase et al., 2021). The eigenvectors corresponding to the larger eigenvalues are found to dominate the transferability of features in adversarial domain adaptation (Chen et al., 2019b). The work of (Chen et al., 2019a) explores the correlation between negative transfer and the spectral components of features and weights in inductive transfer learning. By utilizing the principle of Maximal Coding Rate Reduction (MCR$^2$), it is theoretically proven that the larger several singular values of feature matrix for every class should be equal to learn the maximally diverse representation (Yu et al., 2020; Chan et al., 2020).

Different from these studies, we analyse the connection between adversarial robustness and spectral components of deep features. We aim to explore which components are more fragile under attacks, and propose a method to boost adversarial robustness by constraining spectral properties.

## 3 SPECTRAL ANALYSIS IN FEATURE SPACE

In this section, we investigate the connection between spectral signatures and adversarial robustness. Concretely, we train models by standard training and adversarial training (Madry et al., 2018), and then apply spectral decomposition to attain the spectral signatures. We find that in standard training the eigenvalues reveal a rapidly descending curve while this tendency is alleviated by AT. Then, we find that the adversary tends to add more quantity along eigenvectors with smaller eigenvalues.

### 3.1 CURVE OF EIGENVALUES AND ADVERSARIAL ROBUSTNESS

Given a dataset $D = \{(x_i, y_i)\}_{i=1}^{n}$ including $C$ classes, $x_i$ represents the input data and $y_i$ is the label. DNN is composed of a feature extractor $h(\cdot) : \mathbb{R}^D \to \mathbb{R}^d$ and a linear classifier $g(\cdot) : \mathbb{R}^d \to \mathbb{R}^C$. After centralizing the learned features (*i.e.* $\frac{1}{n}\sum_{i=1}^{n} h(x_i) = 0$), we decompose the learned features by spectral decomposition, which is similar to PCA:

$$\frac{1}{n}\sum_{i=1}^{n} h(x_i) h(x_i)^T = \sum_{j=1}^{d} u_j \lambda_j u_j^T \tag{2}$$

where $\lambda_j$ means the eigenvalues with index $j$ and $u_j \in \mathbb{R}^d$ represents its eigenvector.

Specifically, we train ResNet-18 (He et al., 2016) using both standard training and adversarial training on CIFAR-10. The parameters for AT are the same as (Rice et al., 2020). We calculate the eigenvalues by applying Eq. (2), and the eigenvalues are plotted in Figure 2. The features come from the penultimate layer (512 dimensions). All the features are extracted from the *test set* in CIFAR-10. A part of the eigenvalues is shown for better visualization.

**Difference of models in spectral analysis.** As shown in Figure 2(a)(b), the eigenvalues of a standard trained model drop rapidly at some point, while the sharp decrease of eigenvalues is much alleviated by AT. These sharp spectral signature in standard training makes just a few eigenvalues informative from the opinion of PCA. The model fails to explore the influence of eigenvectors with smaller eigenvalues on classification, so the trained model could not recognize the change of features along eigenvectors with smaller eigenvalues. The eigenvectors which may endow useful features are overly penalized. Consequently, we propose a **hypothesis** that *the severe dominance of the top eigenvectors is a cause of vulnerability in DNN, and the adversary adds more components in eigenvectors with smaller eigenvalues.* We will verify the proposed hypothesis in the next section.

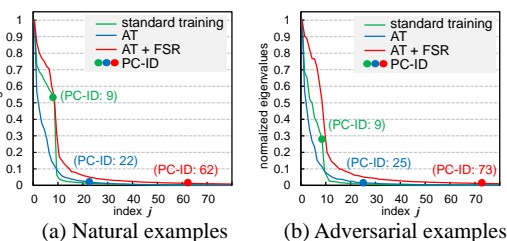

(a) Natural examples    (b) Adversarial examples

Figure 2: Spectral analysis of models with features extracted from (a) natural examples and (b) adversarial examples on CIFAR-10. We scale all the eigenvalues that the largest one is 1. "PC-ID" (Ansuini et al., 2019) denotes the estimated intrinsic dimension (ID) of features. The sharp distribution of eigenvalues in standard trained model leads to a lower ID, while ID becomes higher by imposing adversarial training and FSR.

**Connection with intrinsic dimension.** We introduce intrinsic dimension (ID) to quantitatively describe the decreasing tendency in eigenvalues. ID (i.e., the minimal number of parameters needed to describe a representation) has a close connection with natural accuracy (Ansuini et al., 2019). It has been found that reduction of ID contributes to an improvement on natural accuracy. We adopt PC-ID proposed by (Ansuini et al., 2019) to estimate ID, which is determined by the number of principal components included to describe 90% of the variance. We mark ID in Figure 2 with solid circles. The results reveal the ID of a standard trained model is very small, while models obtained by AT achieve a higher ID, which is contrary to the influence of ID on natural accuracy. This also verifies that there exists a trade-off between generalization and robustness (Tsipras et al., 2019; Zhang et al., 2019) from the perspective of ID. FSR could further increase ID, based on AT.

### 3.2 VARIATION ALONG EIGENVECTORS UNDER ATTACKS

In this section, we aim to verify the hypothesis that adversary adds more components along eigenvectors with smaller eigenvalues in attacking stage. We define a metric to quantitatively observe the change of features along different eigenvectors under attack, called variation in this paper.

**Definition 1 (Alignment)** *Given a dataset $D_s = \{x_{s,i}, y_{s,i}\}_{i=1}^n$ which may be perturbed. The **alignment** of $D_s$ to the pre-given direction $u_j$ is calculated by the expectation over cosine similarity between features extracted by DNN and the direction vector $u_j$:*

$$align(D_s, u_j) = \mathop{\mathbb{E}}_{(x_{s,i}, y_{s,i}) \in D_s} \frac{|\langle h(x_{s,i}), u_j \rangle|}{\|h(x_{s,i})\| \cdot \|u_j\|} \tag{3}$$

*where the norm $\|\cdot\|$ used is Euclidian norm, and $u_j$ is calculated by Eq. (2). The calculation of $u_j$ is based on features covariance of natural samples, and $u_j$ is fixed.*

**Definition 2 (Variation)** *Given a dataset $D$ consist of natural examples and its perturbed dataset $D_{adv}$. The **variation** on direction $u_j$ is defined as the ratio between **alignment** on $D_{adv}$ and $D$:*

$$r(D_{adv}, D, u_j) = \frac{align(D_{adv}, u_j)}{align(D, u_j)} \tag{4}$$

The alignment is correlated with the distance between subspace spanned by $u_j$ and the actual feature space, so the change of alignment under attack is suitable to describe the influence of attacks on direction $u_j$, which is called as **variation**. Our metric is similar to (Tran et al., 2018; Hayase et al., 2021) used for analyzing backdoors, but we define the alignment by cosine similarity while the latter uses the inner product. Compared with inner product, *cosine similarity could eliminate the influence of scale*. In Appendix B.1, we provide a complete procedure to calculate the metrics.

We give an intuitive explanation on why the defined variation is suitable to capture the change of features along different eigenvectors under attacks. An toy model for illustration is shown in Figure 3. Suppose the distribution transfers from Figure 3(a) to (b) under attack, we draw an original data $x$ and its shifted data $x_{adv}$. Take the fixed direction $u_2$ as an illustration. The cosine similarity between $x$ and $u_2$ is equal to $\cos(\theta)$, *i.e.*, $\cos(\theta) = \langle x, u_2 \rangle / \|x\| \cdot \|u_2\|$, which is called as alignment above. If the distribution moves from Figure 3(a) to (b), then the cosine similarity $\cos(\theta)$ increases. Consequently, the change of the cosine similarity can describe how the features change along a direction under attacks. For the vari-

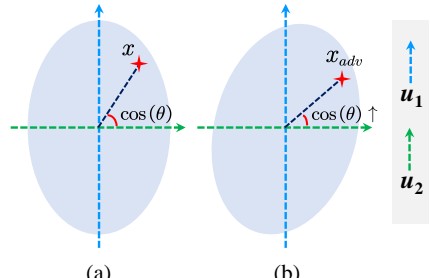

Figure 3: A toy model that demonstrates the validity of the defined variation.

ation defined in Eq. (4), $r(D_{adv}, D, u_j) > 1$ means the features add more components along direction $u_j$, and vice versa. We could compare $r(D, D, u_j)$ with 1 to observe whether the adversary adds or reduces the components along various eigenvectors.

**The adversary adds more components along the eigenvectors with smaller eigenvalues.** We visualize the variation of features to different eigenvectors $\{u_1, \cdots, u_{512}\}$. The results of CIFAR-10 and SVHN are shown in Figure 4. The attacks used include FGSM (Goodfellow et al., 2015),

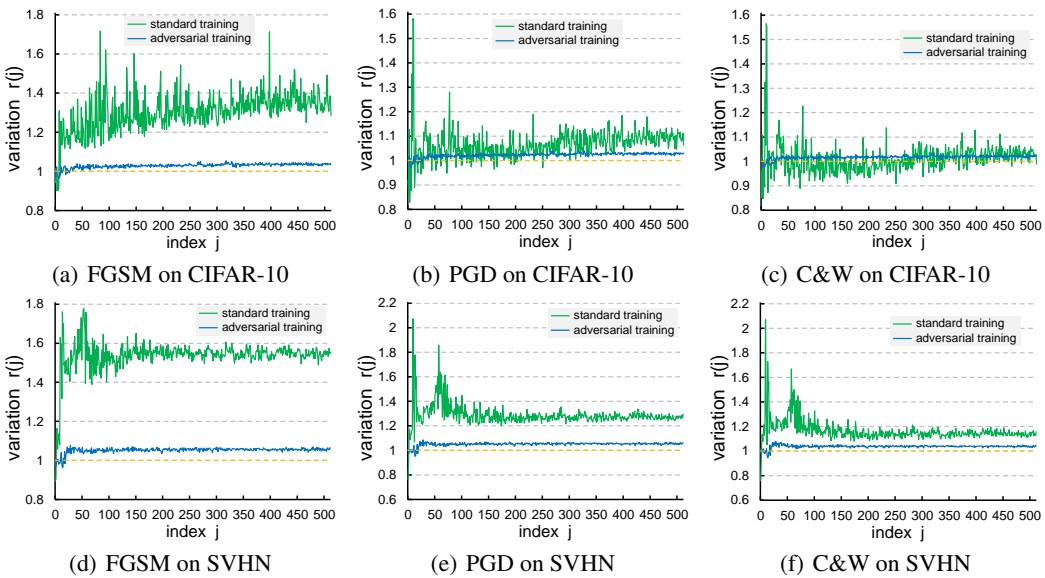

Figure 4: Variation on different eigenvectors in feature space under various attacks. The results reveal that the adversary adds more components along the eigenvectors with smaller eigenvalues.

PGD (step size $\epsilon/10$ for 10 steps) (Madry et al., 2018) and C&W attack (Carlini & Wagner, 2017). We set attack budget $\epsilon = 4/255$ constrained by $\ell_\infty$ norm. As observed in Figure 4(a), the variation keeps close or smaller than 1 for the several largest eigenvalues in standard trained model. However, the variation of smaller eigenvalues is much larger than 1. This means that FGSM tends to add more components along the eigenvectors $u_j$ with smaller eigenvalues in standard training. A similar phenomenon also exists in PGD and C&W, which verifies the hypothesis in Section 3.1. We also provide the curve while changing budget $\epsilon$ and the variation on CIAFR-100 in Appendix B.2.

For models trained by AT, variation of all eigenvectors keeps close to 1, which is distinctly different from standard training. The high variation of directions with smaller eigenvalues visibly decreases. *It is worth noticing that we apply cosine similarity for definition, so the influence from the scale of features is eliminated.* Therefore, the components corresponding to the larger eigenvalues are more robust while the smaller ones are more fragile. Adopting the opinion of robust features (Ilyas et al., 2019), the features along the eigenvectors with larger eigenvalues are regarded as robust features, and these along the direction with smaller eigenvalues are non-robust features. The analysis above motivates us to regularize the spectrum signatures to alleviate the rapid decreasing tendency. If we alleviate the dominance of the top eigenvalues, the adversarial robustness is improved.

## 4 FEATURE SPECTRAL REGULARIZATION

In this section, we propose a method to regularize the eigenvalues, aiming to alleviate the dominance of the top eigenvalues. We first propose our method FSR, and then provide a theoretical analysis.

### 4.1 REALIZATION OF FSR

Through the detailed analysis above, the sharp distribution of eigenvalues weakens the information contained by the smaller eigenvalues, and causes fragility to adversarial attacks. A promising approach to alleviate such phenomenon is to suppressing the largest eigenvalues, which could mitigate the dominance of the largest eigenvalues relatively. Another straightforward idea is to increase the small eigenvalues during training. However, small eigenvalues are usually numerically unstable, which may be easily affected by noise or round-off error in optimization. In this paper, we propose a method called **Feature Spectral Regularization (FSR)** by penalizing the largest eigenvalues of feature covariance in a batch of data:

$$L_{FSR}(F) = w(\tau) \cdot \lambda_{\max}\left(\left(F - \frac{1}{m}\mathbf{1}F\right)^T \left(F - \frac{1}{m}\mathbf{1}F\right)\right) \tag{5}$$

where $\lambda_{\max}(\cdot)$ means the largest eigenvalue of a matrix, $w(\tau) = \min\{\tau/T_0, 1\}$ is the weighting function of current epoch $\tau$ with a hyper-parameter $T_0$, and $\mathbf{1} \in \mathbb{R}^{m \times m}$ is a constant matrix with each element equal to 1 for calculating the mean. Moreover, $F \in \mathbb{R}^{m \times d}$ is the feature matrix of a batch composed of row vectors $h(x_i)^T$, i.e., $F = [h(x_1)|h(x_2)|\cdots|h(x_m)]^T$, and $m$ is the number of samples in a batch of data.

In practice, we can just access the batch of data to approximate the statistics of feature space. However, the eigenvalues change drastically in the early stages of training, which may cause instability on optimization, so we apply a piecewise linear function $w(\tau)$ to smooth the training stage. The $\lambda_{\max}(\cdot)$ in Eq. (5) is equal to the square of the largest singular value in feature matrix $(F - \frac{1}{m}\mathbf{1}F)$ by Singular Value Decomposition (SVD). So the realization of FSR is very simple with the help of Pytorch (Paszke et al., 2019) in a few lines of code.

**Summarized objective loss.** Adversarial training (Madry et al., 2018) has been widely proven to be a strong baseline in adversarial defense. We build and incorporate the proposed FSR into AT training framework for further improving model robustness significantly. The final objective is:

$$L_{adv}(x, y; \theta) = L_{CE}(x_{adv}, y; \theta) + \beta_{FSR} \cdot L_{FSR}(F_{adv}) \tag{6}$$

where $\beta_{FSR}$ is a hyper-parameter that controls the trade-off between two items, $x_{adv}$ represents the adversarial example generated from $x$ by PGD (Madry et al., 2018) using cross-entropy loss, $F_{adv}$ is the feature matrix of adversarial examples. Besides AT, we also apply FSR to defense methods such as TRADES (Zhang et al., 2019) and AT-AWP (Wu et al., 2020). We provide a pseudo-code of our method in Algorithm 1 in Appendix A.

**Computational complexity.** SVD is an extra computational cost induced by FSR. For a matrix $F \in \mathbb{R}^{m \times d}$, the time complexity of SVD is $O\left(\min\left\{m^2 \cdot d, m \cdot d^2\right\}\right)$. Since the batch size is often small, the excess cost in computation is negligible compared to adversarial training.

## 4.2 THEORETICAL ANALYSIS

Consider a linear regression model $\hat{y} = \langle z, \theta \rangle$ with $\ell_2$ perturbation $\delta$ based on the feature $z \in \mathbb{R}^d$, **the underlying parameter obtained by minimizing mean square error** is $\theta_0 \in \mathbb{R}^d$ without perturbation. We assume the mean of features $\mathbb{E}(z) = 0$ and covariance matrix $\text{Var}(z) = \Sigma$. Following the adversarial risk define by (Xing et al., 2021), $\mathcal{R}_{adv}$ is expressed in Eq. (7):

$$\mathcal{R}_{adv}(\theta, \delta) = \mathbb{E}_z \max_{\|z_{adv} - z\| \le \delta} \left(\langle z_{adv}, \theta \rangle - \langle z, \theta_0 \rangle\right)^2. \tag{7}$$

The optimal solution of Eq.(7) denoted as $\theta_{adv}$ has an similar formulation with the ridge regression:

$$\theta_{adv} = (\Sigma + \lambda I)^{-1} \Sigma \theta_0, \tag{8}$$

where $\lambda$ can be regarded as a constant (El Ghaoui & Lebret, 1997; Xing et al., 2021).

Given samples $\{z_i, y_i\}_{i=1}^n$, the feature matrix composed of row vector $z_i^T$ is denoted as $Z \in \mathbb{R}^{n \times d}$, then $\Sigma = \frac{1}{n} Z^T Z$. $y = [y_1, y_2, \cdots, y_n]^T \in \mathbb{R}^n$ is a column vector composed of real-valued output $y_i (i = 1, \cdots, n)$. Previous methods tend to regularize the classifier $\theta$ (El Ghaoui & Lebret, 1997; Xu et al., 2008). However, we could directly analyse how the feature $Z$ influences the robustness of model. If the feature is more robust for classifier, then whether using the classifier $\theta_0$ or $\theta_{adv}$ should have the same prediction. Thus, we define the *residual risk* induced by features $Z$:

$$\min_Z \mathcal{R}_{res}(Z) = \|Z\theta_{adv} - Z\theta_0\|_2, \quad s.t. \|Z\|_F^2 = s_0. \tag{9}$$

It is essential to normalize the scale of features for comparable representation, so we restrict the norm of $Z$ as used in (Yu et al., 2020). The SVD of matrix $Z \in \mathbb{R}^{n \times d}$ has the form: $Z = UDV^T$. Here $U = [u_1, \cdots, u_n] \in \mathbb{R}^{n \times n}$ and $V \in \mathbb{R}^{d \times d}$ are orthogonal matrices. Observe that the constraint for $Z$ only depends on its singular values, i.e., $\|Z\|_F^2 = \sum_{i=1}^{\min\{n,d\}} \sigma_i^2 = s_0$. Suppose $d < n$ and $\sigma_i > 0$, then $\mathcal{R}_{res}$ is simplified to the expression by combining Eq. (9) and Eq. (8):

$$\min_{(\sigma_1, \cdots, \sigma_d)} \mathcal{R}_{res}(\sigma_1, \cdots, \sigma_d) = \min_{(\sigma_1, \cdots, \sigma_d)} \left\| \sum_{j=1}^d u_j \frac{\lambda n}{\sigma_j^2 + \lambda n} u_j^T y \right\|_2, \quad s.t. \sum_{j=1}^d \sigma_j^2 = s_0. \tag{10}$$

Table 1: Test accuracy (%) on CIFAR-10 under white-box/black-box attacks using ResNet-18. The maximum $\ell_\infty$ perturbation is $\epsilon = 8/255$. The best results are boldfaced for highlight.

| Defense | White-box Attack | | | | Black-box Attack | | | |
|---|---|---|---|---|---|---|---|---|
| | Natural | FGSM | PGD-20 | $CW_\infty$ | FGSM | PGD-20 | $CW_\infty$ | AA |
| AT | 82.02 | 57.08 | 51.63 | 50.28 | 63.63 | 61.12 | 61.36 | 48.13 |
| AT + FSR | **82.18** | **57.45** | **52.24** | **51.25** | **63.74** | **61.25** | **61.63** | **48.71** |
| TRADES | 83.80 | 57.93 | 51.02 | 49.53 | 65.34 | 62.94 | 62.68 | 48.12 |
| TRADES + FSR | **84.64** | **58.54** | **51.66** | **50.00** | **65.98** | **63.60** | **63.40** | **48.66** |

Table 2: Test accuracy (%) on CIFAR-100 under white-box/black-box attacks using ResNet-18. The maximum $\ell_\infty$ perturbation is $\epsilon = 8/255$.

| Defense | White-box Attack | | | | Black-box Attack | | | |
|---|---|---|---|---|---|---|---|---|
| | Natural | FGSM | PGD-20 | $CW_\infty$ | FGSM | PGD-20 | $CW_\infty$ | AA |
| AT | **55.57** | 30.86 | 27.62 | 25.93 | 41.50 | 40.94 | 42.27 | 23.74 |
| AT + FSR | 54.57 | **32.16** | **29.18** | **26.90** | **41.93** | **41.42** | **42.69** | **24.79** |
| TRADES | 55.38 | 29.77 | 26.31 | 23.58 | 40.76 | 39.98 | 41.31 | 22.65 |
| TRADES + FSR | **57.55** | **32.16** | **28.49** | **24.99** | **42.57** | **41.91** | **43.22** | **23.82** |

Table 3: Test accuracy (%) on SVHN under white-box/black-box attacks using ResNet-18. The maximum $\ell_\infty$ perturbation is $\epsilon = 8/255$.

| Defense | White-box Attack | | | | Black-box Attack | | | |
|---|---|---|---|---|---|---|---|---|
| | Natural | FGSM | PGD-20 | $CW_\infty$ | FGSM | PGD-20 | $CW_\infty$ | AA |
| AT | **89.95** | **59.41** | 47.68 | 44.84 | 67.24 | 62.09 | 63.57 | 41.83 |
| AT + FSR | 88.56 | 59.02 | **51.17** | **47.72** | **67.39** | **62.72** | **64.54** | **44.14** |
| TRADES | 92.42 | 68.60 | 58.27 | 55.31 | 73.13 | 68.90 | 70.11 | 52.54 |
| TRADES + FSR | **92.69** | **69.82** | **59.22** | **55.69** | **73.35** | **69.32** | **70.60** | **52.79** |

**Theorem 1** $\mathcal{R}_{res}(\sigma_1, \cdots, \sigma_d)$ *is minimum when all the singular values of $Z$ are equal.*

As FSR penalizes the largest singular value in feature matrix, it alleviates the dominance of large singular values and helps contribute to equal singular values under the normalized condition, thus helping reduce the residual risk induced by features. Proof of Theorem 1 is provided in Appendix C.

## 5 EXPERIMENTS

In this section, we evaluate the effectiveness of the proposed FSR on CIFAR-10 (Krizhevsky et al., 2009), CIFAR-100 (Krizhevsky et al., 2009) and SVHN (Netzer et al., 2011). FSR is applied to two baselines: 1) AT (Madry et al., 2018; Rice et al., 2020); 2) TRADES (Zhang et al., 2019). It is worth noticing that AT is the most effective method to improve adversarial robustness (Rice et al., 2020; Pang et al., 2021; Gowal et al., 2021) in RobustBench (Croce et al., 2020).

**Experimental Settings.** For CIFAR-10 and CIFAR-100, we set the $\ell_\infty$ perturbation with $\epsilon = 8/255$, the step size of attack $2/255$, and the inner iteration steps 10. The step size is $1/255$ for SVHN. We train ResNet-18 (He et al., 2016) using momentum optimizer with the initial learning rate of 0.1. For AT, we train 200 epochs and the learning rate decays with a factor of 0.1 at 100 and 150 epochs (Rice et al., 2020). For TRADES, we train 120 epochs with the learning rate divided by 0.1 at epochs 75, 90, and 100 (Zhang et al., 2019). The parameter for regularization $(1/\lambda)$ is set as 4 for TRADES. For FSR, we set $\beta_{FSR} = 0.01$. Other hyper-parameters keep the same as their original paper. Considering that the settings have a distinct influence on robustness (Pang et al., 2021; Gowal et al., 2021), the hyper-parameters remain unchanged while adding our FSR.

### 5.1 PERFORMANCE UNDER WHITE-BOX ATTACKS

To test the adversarial robustness of model in detail, we adopt various white-box adversarial attacks: FGSM (Goodfellow et al., 2015), PGD-20 (step size $\epsilon/4$) (Madry et al., 2018), CW ($\ell_\infty$ version optimized by PGD) (Carlini & Wagner, 2017). Following the instruction proposed by (Carlini et al.,

Table 4: Test accuracy (%) across different datasets based on final checkpoint under white-box attacks using ResNet-18. The maximum $\ell_\infty$ perturbation is $\epsilon = 8/255$.

| Defense | CIFAR-10 | | CIFAR-100 | | SVHN | |
|---|---|---|---|---|---|---|
| | PGD-20 | $CW_\infty$ | PGD-20 | $CW_\infty$ | PGD-20 | $CW_\infty$ |
| AT | 43.65 | 44.17 | 19.94 | 20.46 | 41.94 | 43.35 |
| AT + FSR | **44.91** | **44.79** | **22.46** | **21.12** | **44.04** | **44.25** |
| TRADES | 50.85 | **49.66** | 26.32 | 23.44 | 57.99 | 55.05 |
| TRADES + FSR | **51.15** | 49.54 | **28.43** | **24.68** | **59.11** | **59.13** |

Table 5: Test accuracy (%) on CIFAR-10 under white-box attacks using ResNet-18 based on AWP. We report the results based on best checkpoint. The maximum $\ell_\infty$ perturbation is $\epsilon = 8/255$.

| Defense | Natural | FGSM | PGD-20 | $CW_\infty$ | AA |
|---|---|---|---|---|---|
| AT-AWP | 80.23 | 59.03 | 55.13 | 51.38 | 49.48 |
| AT-AWP + FSR | **80.63** | **59.11** | **55.13** | **51.99** | **49.93** |

2019), we use 5 random starts at random offsets away from the initial when applying iterative attacks. The settings for evaluation promote a quite strong attack. We report the best checkpoint (the highest robustness under PGD from different checkpoints) as used in (Rice et al., 2020). The test accuracy is reported in Table 1~3. The results show that FSR can further improve adversarial robustness, compared with the baseline models. We also test the robustness under AutoAttack (AA) (Croce & Hein, 2020), which is now regarded as the strongest attack to evaluate adversarial robustness. As shown in the table, FSR is effective to improve robustness under AA. The detailed results show that our FSR can indeed boost robustness, rather than depending on obfuscated gradients (Athalye et al., 2018) or gradient masking (Papernot et al., 2017). The improvement on various datasets reveals that our method has a consistent promotion, rather than relying on a special dataset.

The performance of final checkpoint is another essential probe to analyse the adversarial robustness of models. We also report the performance of final checkpoint, as shown in Table 4. The results show that FSR could also improve the performance in final checkpoint on various datasets.

**Combination with other methods.** To validate that FSR is an effective module, we try to combine FSR with Adversarial Weight Perturbation (AWP) (Wu et al., 2020). AWP helps improve adversarial robustness and effectively alleviates the overfitting (Rice et al., 2020) in PGD-based AT. It is one of the strongest defense methods in RobustBench (Croce et al., 2020), which means it is a high baseline. The results are listed in Table 5, and the results show that we could even **attain improvement on adversarial robustness based on the high baseline AWP**, especially under CW attack and AA.

Table 6: Test accuracy (%) on CIFAR-10 under white-box attacks using WideResNet-34-10.

| Defense | FGSM | | PGD-20 | | CW | |
|---|---|---|---|---|---|---|
| | Best | Last | Best | Last | Best | Last |
| AT | 61.85 | 57.69 | **55.10** | 47.46 | 53.54 | 48.12 |
| AT + FSR | **62.22** | **59.12** | 54.93 | **47.89** | **53.98** | **48.32** |
| AT-AWP | 63.64 | **63.96** | 58.17 | 57.09 | 55.81 | 55.11 |
| AT-AWP + FSR | **64.19** | 63.59 | **58.23** | **58.01** | **55.99** | **56.25** |

**Performance under WideResNet.** To explore the performance of our model in large networks, we use WideResNet-34-10 (depth 34 and width 10) (Zagoruyko & Komodakis, 2017) as another backbone network. In this part, we realize our method based on the well-known effective methods: AT (Madry et al., 2018; Rice et al., 2020) and AT-AWP (Wu et al., 2020) on CIFAR-10. All the parameters of baselines are the same as used in their original paper. The settings of threaten model are the same as ResNet-18 in the previous section. We show the results of both best checkpoint and final checkpoint in Table 6. The "Best" means the *checkpoint of best performance under PGD*, and the "Last" is the performance of final checkpoint. The results verify that our method is also effective in WideResNet, and we especially attain great promotion in final checkpoint based on AWP.

## 5.2 PERFORMANCE UNDER BLACK-BOX ATTACKS

Transferability is an intriguing property about adversarial examples, which can be used to implement black-box attack (Papernot et al., 2016a; Liu et al., 2017). Adversarial examples generated by the

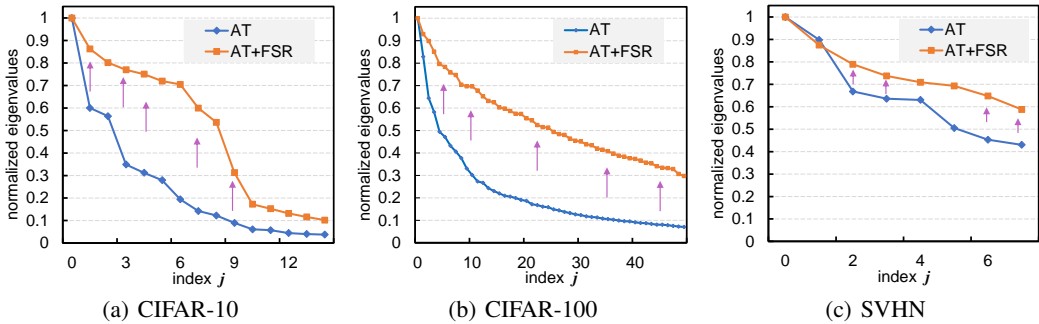

Figure 5: Spectral analysis of FSR. (a) normalized eigenvalues (max-normalized) on CIFAR-10; (b) normalized eigenvalues on CIFAR-100; (c) normalized eigenvalues on SVHN.

substitute model can be used to attack the target model with the weights and architecture unknown to the adversary. Transfer attack is beneficial to verify that our method does not rely on gradient masking (Papernot et al., 2017). As suggested by (Carlini et al., 2019), we adopt another adversarially trained ResNet-18 as the substitute model. The results are shown in Table 1∼ 3, revealing that our methods also improve robustness under black-box attacks.

### 5.3 FURTHER UNDERSTANDING OF FSR

In this part, we analyse the influence of FSR on the eigenvalue spectrum. The results are shown in Figure 5. Our motivation is that the top eigenvalues dominate the distribution of eigenvalues, so we use FSR to increase the overall eigenvalues relatively. To eliminate the influence from the scale of features, the eigenvalues are divided by the maximum eigenvalue. As shown in the figure, FSR increases the eigenvalues relatively, which is consistent with our intention for FSR. In Appendix D, we show that simply using FSR without AT also promotes a mild improvement on robustness.

### 5.4 ABLATION STUDIES

**Sensitivity analysis of $\beta_{FSR}$.** We explore how the weight of FSR $\beta_{FSR}$ influences the performance of models. Following the same settings as the previous section, we train our models on CIFAR-10 using ResNet-18 with the change of $\beta_{FSR}$. The results are reported Table 7. As observed in Table 7, FSR can significantly improve robustness with a wide value range, which verifies the stability and effectiveness of the proposed method. In this paper, we choose the parameter by considering both generalization and robustness, so we choose $\beta_{FSR} = 0.01$. If we focus merely on adversarial robustness, tuning $\beta_{FSR}$ could still promote a further improvement on robustness.

Table 7: Test accuracy (%) under white-box attacks for different $\beta_{FSR}$.

| $\beta_{FSR}$ | 0 | 0.005 | 0.010 | 0.020 | 0.060 |
|---|---|---|---|---|---|
| Natural | 82.02 | 81.96 | 82.18 | 81.84 | 81.74 |
| PGD-20 | 51.63 | 52.42 | 52.24 | 52.77 | 52.21 |
| CW$_\infty$ | 50.28 | 51.04 | 50.77 | 51.21 | 50.75 |

## 6 CONCLUSION

In this paper, we delve into the discrepancy between natural examples and adversarial examples from the perspective of spectral analysis. The variation of different eigenvectors under adversarial attacks is analysed. It is shown that the spectral directions with smaller eigenvalues are more fragile under attack, which is induced by the dominance of the top eigenvectors. Based on the analysis, a method called Feature Spectral Regularization (FSR) is proposed to penalize the largest eigenvalues of the batch covariance matrix, which is numerically stable and could enlarge the overall eigenvalues relatively. We also provide a theoretical explanation in robust linear regression. FSR is a simple module readily pluggable into any defense method. Through comprehensive experiments, we show that FSR can effectively improve adversarial robustness when combined with many defense methods.

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

## A   Algorithm of FSR

---

**Algorithm 1** Robust Training with FSR

---

**Input:** Training data $D = \{(x_i, y_i)\}_{i=1,\cdots,N}$, DNN composed of feature extractor $h(\cdot)$ and linear classifier $g(\cdot)$, maximum training steps $T$

**Output:** Robust model $\{h(\cdot), g(\cdot)\}$

 1: **for** $t$ in $[1, 2, \cdots, T]$ **do**
 2:    Read mini-batch $\{x_1, x_2, \cdots, x_b\}$
 3:    Generate adversarial robustness using PGD
 4:    Compute the FSR loss in Eq. (5) using adversarial examples
 5:    Compute the overall loss in Eq. (6)
 6:    Optimize all parameters of model using gradient descent
 7: **end for**

---

## B   The defined variation on more datasets

In this part, we provide more experiments about the **variation** defined by us. We first provide a detailed algorithm to calculate the variation. More results about the variation are shown following.

### B.1   Algorithm for Calculating the Variation

---

**Algorithm 2** Calculating the Alignment and Variation

---

**Require:** Dataset $D = \{(x_i, y_i)\}_{i=1,\cdots,N}$ (natural examples), the feature extractor $h(\cdot)$.

**Ensure:** Variation to different eigenvectors $r(D_{adv}, D, u_j)$ $(j = 1, \cdots, d)$.

 1: Centralize the features $h(x_i) \leftarrow h(x_i) - \frac{1}{N}\sum_{i=1}^{N} h(x_i)$
 2: Calculate the covariance matrix $\frac{1}{N}\sum_{i=1}^{N} h(x_i)h(x_i)^T$
 3: Apply eigendecomposition to the covariance matrix to get the eigenvalues and eigenvectors
      $\{\lambda_1, \cdots, \lambda_d\}, \{u_1, \cdots, u_d\} = \mathrm{eig}\left(\frac{1}{N}\sum_{i=1}^{N} h(x_i)h(x_i)^T\right)$
 4: Generate the adversarial example $x_{adv,i}$ from $x_i$ by attack $(i = 1, \cdots, N)$
 5: Centralize the features of adversarial examples $h(x_{adv,i}) \leftarrow h(x_{adv,i}) - \frac{1}{N}\sum_{i=1}^{N} h(x_{adv,i})$
 6: **for** $j = 1, \cdots, d$ **do**
 7:    Calculate the cosine similarity between features and eigenvectors
      $align(D, u_j) = \frac{1}{N}\sum_{i=1}^{N} |\langle h(x_i), u_j\rangle| / \|h(x_i)\| \cdot \|u_j\|$
      $align(D_{adv}, u_j) = \frac{1}{N}\sum_{i=1}^{N} |\langle h(x_{adv,i}), u_j\rangle| / \|h(x_{adv,i})\| \cdot \|u_j\|$
 8:    Calculate the variation to different eigenvectors
      $r(D_{adv}, D, u_j) = align(D_{adv}, u_j) / align(D, u_j)$
 9: **end for**

---

### B.2   More experiments about the defined variation

In this part, we show more results about variation to eigenvectors on CIFAR-10, CIFAR-100 and SVHN. We compare the models trained on natural examples and adversarial examples. The adversarially trained models adopt the PGD-AT Madry et al. (2018) with $\ell_\infty$ perturbation 8/255, and the parameters are the same as Rice et al. (2020). From the curve shown in Figure 6∼8, we can see that the variation on CIFAR-100 and SVHN reveals the same tendency as CIFAR-10, i.e., adversarial attacks tend to increase the components along the eigenvectors with smaller eigenvalues.

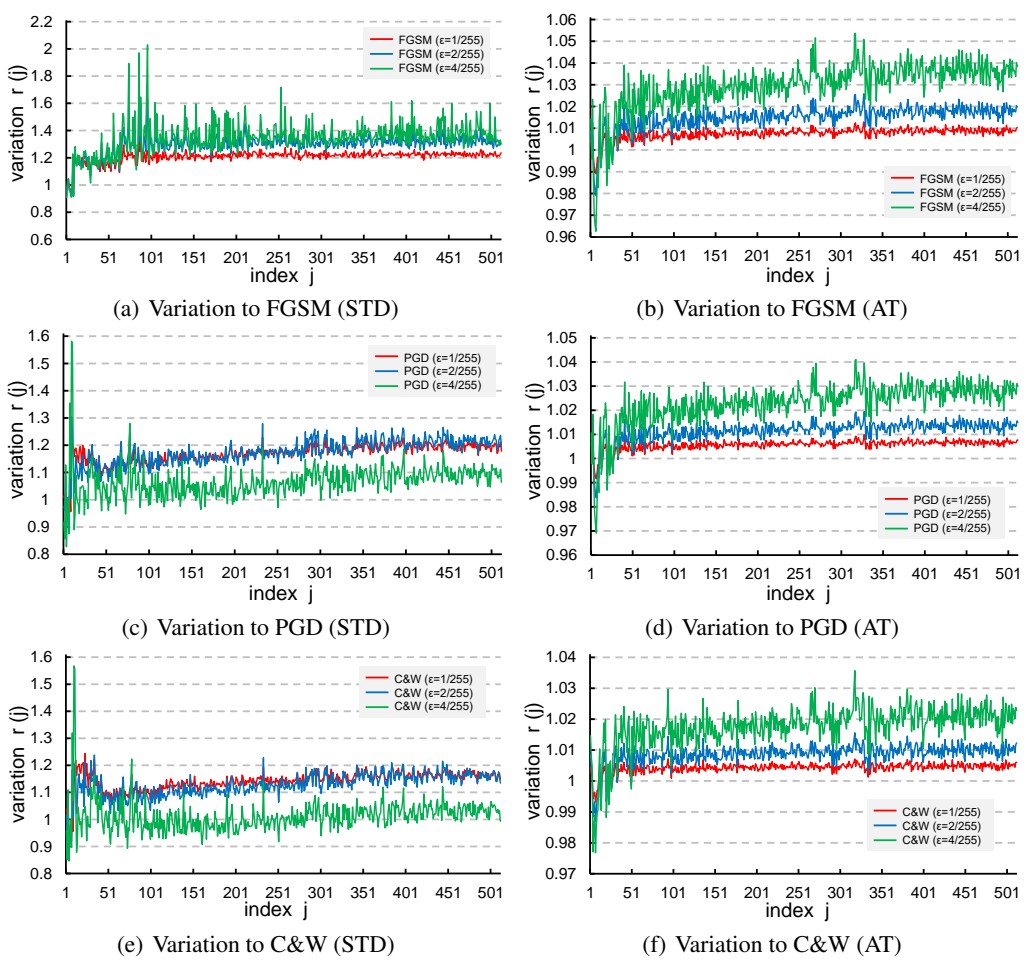

Figure 6: Variation of all eigenvectors in feature space to adversarial attacks on CIFAR-10. 'STD' means the model is trained on natural examples. 'AT' means the model is trained on adversarial examples. It should be noticed that the range of ordinate values for 'STD' is different from 'AT'.

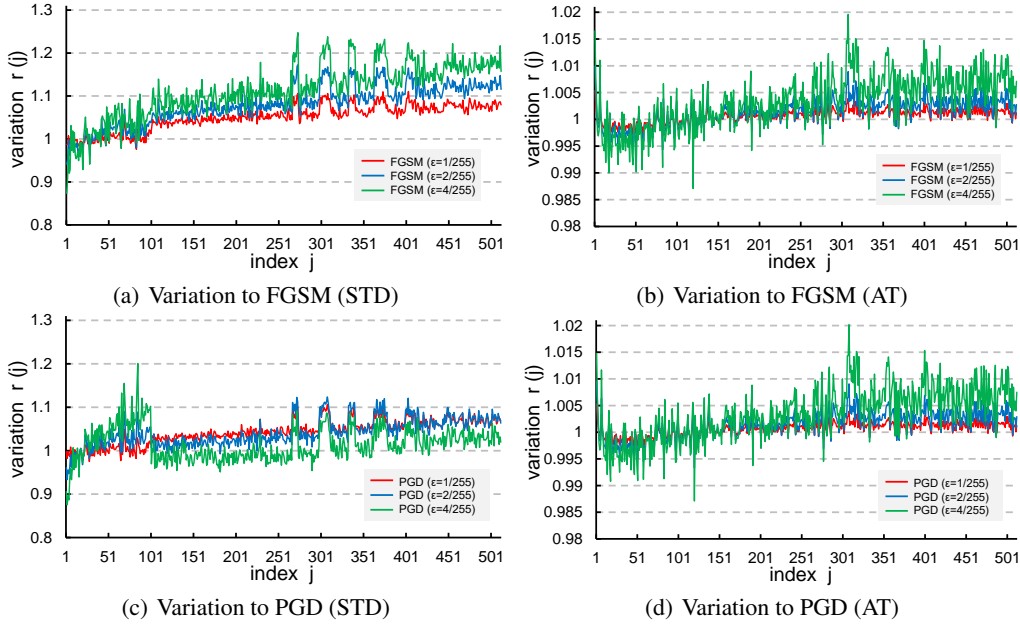

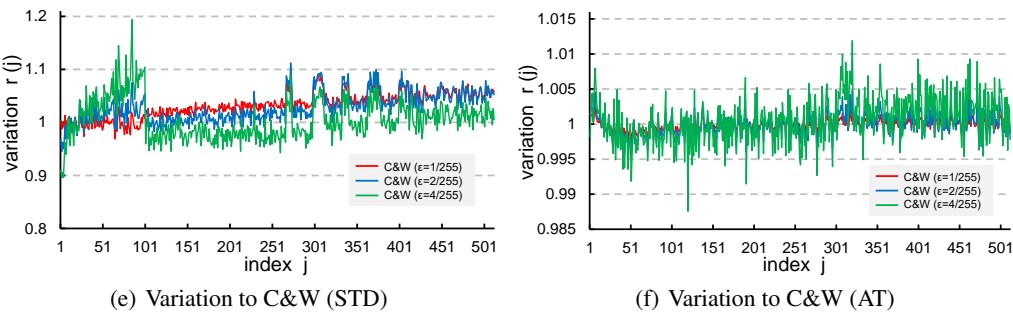

(e) Variation to C&W (STD)

(f) Variation to C&W (AT)

Figure 7: Variation of all eigenvectors in feature space to adversarial attacks on CIFAR-100.

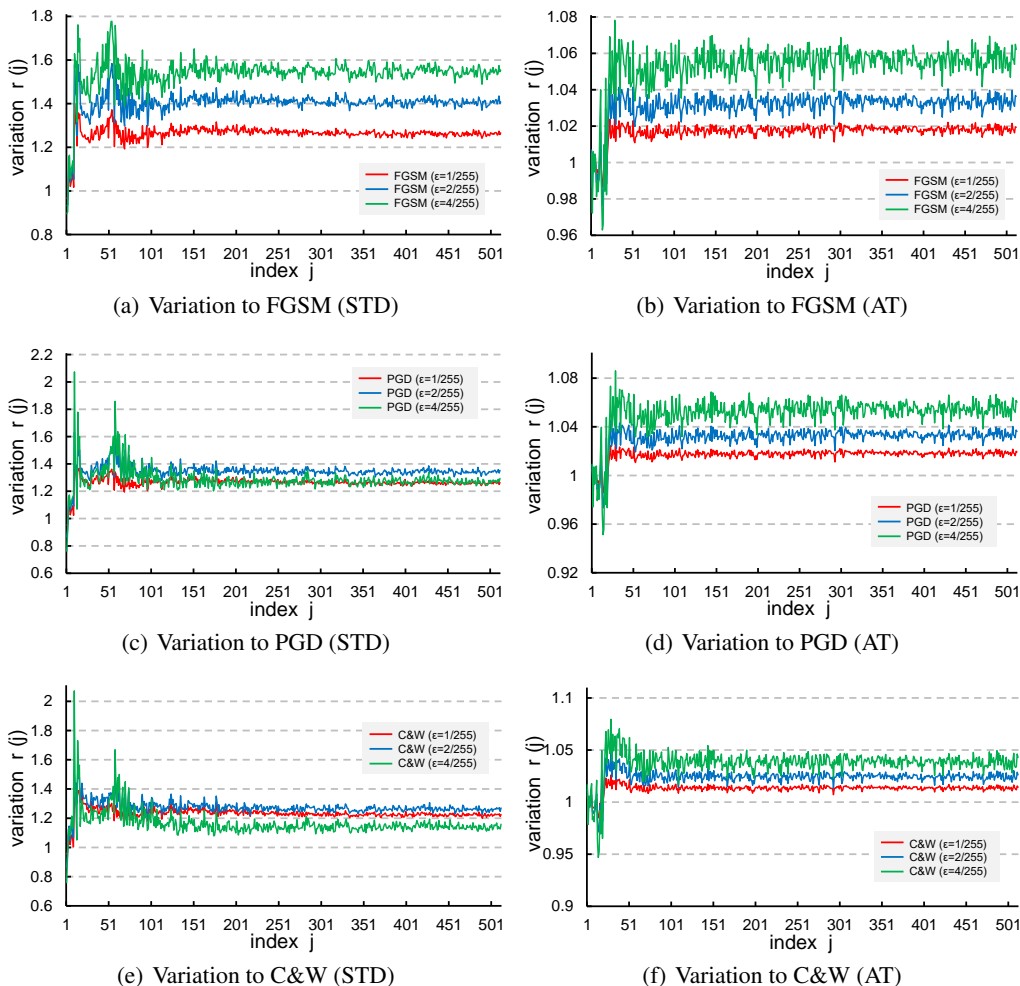

(a) Variation to FGSM (STD)

(b) Variation to FGSM (AT)

(c) Variation to PGD (STD)

(d) Variation to PGD (AT)

(e) Variation to C&W (STD)

(f) Variation to C&W (AT)

Figure 8: Variation of all eigenvectors in feature space to adversarial attacks on SVHN.

## C  PROOF

In this part, we provide the proof of Theorem 1 in Section 4.2.

Consider the linear regression model,

$$\hat{y} = \langle z, \theta \rangle, \tag{11}$$

where $z \in \mathbb{R}^d$ means the feature, and $\theta \in \mathbb{R}^d$ represents the parameter of model. We assume that the features have zero mean, and the covariance is denoted as $\Sigma$, *i.e.*, $\mathbb{E}(z) = 0, \mathrm{Var}(z) = \Sigma$.

Given samples $\{z_i, y_i\}_{i=1}^n$, the feature matrix composed of row vectors $z_i^T$ is represented as feature matrix $Z \in \mathbb{R}^{n \times d}$, then $\Sigma = \frac{1}{n} Z^T Z$. Assuming that $Z$ has full column rank, if we *minimize the residual sum of squares* in Eq. (11), then the solution is:

$$\theta_0 = \left(Z^T Z\right)^{-1} Z^T y, \tag{12}$$

where $y \in \mathbb{R}^n$ means the real-valued output, *i.e.*, $y^T = (y_1, y_2, \cdots, y_n)$.

When the feature $z$ is noised by $\ell_2$ perturbation $\delta$, we can define the adversarial risk induced by the perturbation as (Xing et al., 2021):

$$\mathcal{R}_{adv}\left(\theta, \delta\right) = \mathbb{E}_z \max_{\|z_{adv} - z\| \leq \delta} \left(\langle z_{adv}, \theta \rangle - \langle z, \theta_0 \rangle\right)^2. \tag{13}$$

The optimal solution to Eq. (13) denoted as $\theta_{adv}$ is

$$\theta_{adv} = \left(\Sigma + \lambda I\right)^{-1} \Sigma \theta_0, \tag{14}$$

where $\lambda$ can be regarded as a constant. The work of (El Ghaoui & Lebret, 1997; Xing et al., 2021) provides more details about how to get $\lambda$.

A better feature should lead to the same prediction whether using classifier $\theta_0$ or $\theta_{adv}$, so we define the *residual risk* induced by features

$$\min_Z \mathcal{R}_{res}\left(Z\right) = \min_Z \|Z\theta_{adv} - Z\theta_0\|_2, \quad s.t. \|Z\|_F^2 = s_0. \tag{15}$$

We normalize the scale of features by Frobenius norm to avoid the influence of scale (Yu et al., 2020), which is just correlated to its singular values $\{\sigma_1, \cdots, \sigma_d\}$:

$$\|Z\|_F^2 = \text{trace}\left(Z^T Z\right) = \sum_{j=1}^d \sigma_j^2 = s_0. \tag{16}$$

The SVD of the $n \times d$ matrix $Z$ has the form:

$$Z = UDV^T, \tag{17}$$

where $U = [u_1, \cdots, u_n] \in \mathbb{R}^{n \times n}$ and $V \in \mathbb{R}^{d \times d}$ are orthogonal matrices. $D = \begin{bmatrix} D_0 \\ 0 \end{bmatrix} \in \mathbb{R}^{n \times d}$ and $D_0 = \text{diag}\left(\sigma_1, \cdots, \sigma_d\right), \sigma_1 \geq \sigma_2 \geq \cdots \geq \sigma_d$.

After applying SVD, we can simplify the Eq. (15) to following expression:

$$\min_{(\sigma_1, \cdots, \sigma_d)} \mathcal{R}_{res}\left(\sigma_1, \cdots, \sigma_d\right) = \min_{(\sigma_1, \cdots, \sigma_d)} \left\| \sum_{j=1}^d u_j \frac{\lambda n}{\sigma_j^2 + \lambda n} u_j^T y \right\|_2, \quad s.t. \sum_{j=1}^d \sigma_j^2 = s_0. \tag{18}$$

According to the property of the subordinate matrix norm, we could derive an upper bound of $\mathcal{R}_{res}$:

$$\left\| \sum_{j=1}^d u_j \frac{\lambda n}{\sigma_j^2 + \lambda n} u_j^T y \right\|_2 \leq \left\| \sum_{j=1}^d u_j \frac{\lambda n}{\sigma_j^2 + \lambda n} u_j^T \right\|_2 \cdot \|y\|_2. \tag{19}$$

Consider that $y$ can be arbitrary value, so the minimization of $\mathcal{R}_{res}$ requires that the first item to the right of the inequality sign in Eq. (19) is as small as possible, i,e,

$$\min_{(\sigma_1, \cdots, \sigma_d)} \left\| \sum_{j=1}^d u_j \frac{\lambda n}{\sigma_j^2 + \lambda n} u_j^T \right\|_2 = \min_{(\sigma_1, \cdots, \sigma_d)} \frac{\lambda n}{\sigma_d^2 + \lambda n}. \tag{20}$$

The minimization of $\mathcal{R}_{res}$ is simplified to the following expression:

$$\arg\min_{(\sigma_1, \cdots, \sigma_d)} \mathcal{R}_{res}\left(\sigma_1, \cdots, \sigma_d\right) = \arg\min_{(\sigma_1, \cdots, \sigma_d)} \frac{\lambda n}{\sigma_d^2 + \lambda n}, \quad s.t. \sum_{j=1}^d \sigma_j^2 = s_0. \tag{21}$$

Since $\sigma_1 \geq \sigma_2 \geq \cdots \geq \sigma_d$, the optimal solution of Eq. (21) is that all singular values are equal.

## D  FSR IMPROVES CONFIDENCE CALIBRATION AND ROBUSTNESS

We verify that network trained using only FSR without AT attains a *mild improvement* on adversarial robustness. We test the performance of model under FGSM, and the results are listed in Table 8.

Table 8: Test accuracy (%) under FGSM on CIFAR-10 using ResNet-18.

| Attack Strength | 1/255 | 2/255 | 3/255 | 4/255 | 5/255 |
|---|---|---|---|---|---|
| ERM | 65.74 | 54.61 | 49.55 | 46.96 | 44.69 |
| ERM + FSR | **70.01** | **59.19** | **53.68** | **49.71** | **46.61** |

In addition to robustness, another important but undesirable aspect of DNN is reliability. When using classifiers for prediction, the predictive confidence should be indicative of the actual likelihood of correctness (Guo et al., 2017). Confidence calibration focuses on the mismatch problem between a model's confidence and its correctness. A popular metric used to measure calibration is the Expected Calibration Error (ECE) (Naeini et al., 2015; Guo et al., 2017), defined as the expected absolute difference between the models' confidence and its accuracy. We explore the efficacy of FSR on confidence calibration.

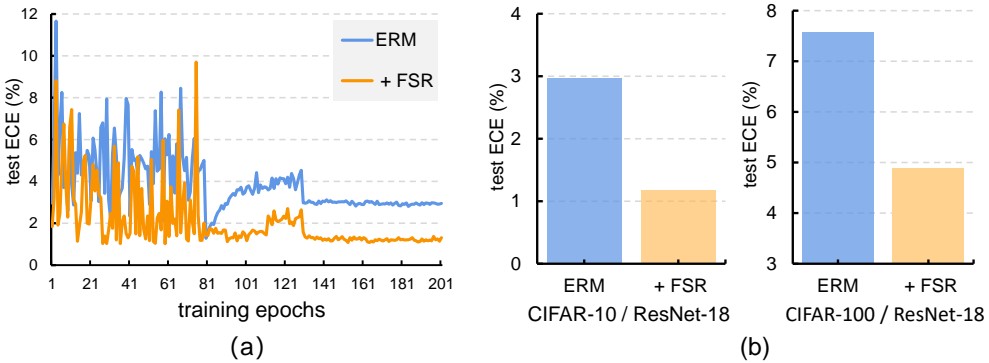

Figure 9: (a) Comparison of test ECE (lower is better) curves when training ResNet-18 on CIFAR-10. (b) FSR can improve the confidence calibration.

As shown in Figure 9, for the baseline, there is a marked rise in the test ECE at the later period of the training process, which indicates the problem of overconfidence. Remarkably, FSR has an obvious improvement on confidence calibration compared with the baseline, *e.g.*, reduces the ECE value from 2.97% to 1.18% for CIFAR-10.

## E  FURTHER EXPERIMENTS ABOUT ADVERSARIAL ROBUSTNESS

In this part, we add two experiments about the proposed FSR on adversarial robustness. We first report the results (i.e., mean and standard deviation) when all experiments in Table 1∼3 are repeated five times. Then, the test accuracy of suppressing more eigenvalues is also provided.

### E.1  EXPERIMENTS ABOUT STANDARD DEVIATION

We repeat all the experiments in Table 1∼3 five times, and report the results under white-box attack in following Table. The performance in Table 9∼11 verifies that our proposed method achieves a steady improvement.

Table 9: Test accuracy (%) on CIFAR-10 under white-box attacks using ResNet-18. The maximum $\ell_\infty$ perturbation is $\epsilon = 8/255$.

| Defense | Natural | FGSM | PGD-20 | C&W | AA |
|---|---|---|---|---|---|
| AT | 82.14±0.24 | 57.38±0.37 | 51.52±0.19 | 50.52±0.28 | 48.07±0.14 |
| AT + FSR | **82.57**±0.36 | **58.02**±0.52 | **52.12**±0.16 | **51.36**±0.17 | **48.91**±0.17 |
| TRADES | 83.73±0.06 | 58.09±0.14 | 51.10±0.10 | 49.67±0.13 | 48.18±0.13 |
| TRADES + FSR | **84.08**±0.48 | **58.43**±0.10 | **51.66**±0.05 | **50.00**±0.08 | **48.62**±0.17 |

Table 10: Test accuracy (%) on CIFAR-100 under white-box attacks using ResNet-18. The maximum $\ell_\infty$ perturbation is $\epsilon = 8/255$.

| Defense | Natural | FGSM | PGD-20 | C&W | AA |
|---|---|---|---|---|---|
| AT | **55.63**±0.06 | 30.87±0.12 | 27.62±0.20 | 26.14±0.18 | 23.93±0.17 |
| AT + FSR | 54.58±0.05 | **32.16**±0.14 | **29.01**±0.26 | **26.88**±0.37 | **24.76**±0.14 |
| TRADES | 56.03±0.65 | 29.84±0.07 | 26.22±0.25 | 23.83±0.32 | 22.81±0.20 |
| TRADES + FSR | **57.64**±0.16 | **32.10**±0.08 | **28.38**±0.15 | **25.14**±0.21 | **23.85**±0.17 |

Table 11: Test accuracy (%) on SVHN under white-box attacks using ResNet-18. The maximum $\ell_\infty$ perturbation is $\epsilon = 8/255$.

| Defense | Natural | FGSM | PGD-20 | C&W | AA |
|---|---|---|---|---|---|
| AT | **90.16**±0.21 | **59.94**±0.47 | 47.86±0.21 | 45.24±0.38 | 42.00±0.20 |
| AT + FSR | 89.37±0.75 | 59.79±0.77 | **50.86**±0.38 | **47.85**±0.24 | **44.41**±0.24 |
| TRADES | **92.48**±0.11 | 68.73±0.15 | 58.82±0.34 | 55.48±0.21 | 52.56±0.06 |
| TRADES + FSR | 92.39±0.35 | **69.72**±0.29 | **59.04**±0.22 | **55.64**±0.07 | **52.78**±0.28 |

### E.2 SUPPRESSING MORE EIGENVALUES

In the main body of our paper, the proposed FSR only penalizes the largest eigenvalue. In this part, we explore whether penalizing more eigenvalues could achieve a better performance under attack. The results are listed in Table 12. As we properly increase the number of suppressed eigenvalues $k$, the adversarial robustness of models could be further improved. However, if we keep increasing $k$, the robustness of the model declines. We think it is due to that $k$ has some trade-off with the weight of FSR $\beta_{\text{FSR}}$, i.e., a larger $k$ may require a smaller $\beta_{\text{FSR}}$.

Table 12: Test accuracy (%) on CIFAR-10 under AutoAttack (AA) using ResNet-18 while suppressing the largest $k$ eigenvalues. The maximum $\ell_\infty$ perturbation is $\epsilon = 8/255$.

| $k$ | 1 | 2 | 4 | 8 | 12 | 16 |
|---|---|---|---|---|---|---|
| AA | 48.71 | 48.95 | 49.02 | **49.17** | 48.55 | 48.14 |

## F MORE DISCUSSIONS BETWEEN SPECTRAL SIGNATURES AND ROBUSTNESS

In this section, we discuss the connection between our explanation of adversarial examples in the paper and previous studies. We mainly talk about the connection with interpretations form the perspective of Lipschitz constant and manifold.

### F.1 CONNECTION BETWEEN FSR AND LIPSCHITZ CONSTANT

In this part, we first rederive some basic conclusions about Lipschitz constant in DNN as illustrated in (Yoshida & Miyato, 2017). For a feature extractor $g(x) : \mathbb{R}^D \to \mathbb{R}^d$ composed of $L$ layers, the weights of each layer in the network are denoted as $\{W^l, b^l\}_{l-1}^L$. Then the feature extractor $g(x) = W^L \phi^{L-1} \left( W^{L-1} \phi^{L-2} (\cdots) + b^{L-1} \right) + b^L$, where $\phi^l$ represents the activation function. We further assume that each activation function is a ReLU. In this case, $\phi^l$ act as input-dependent diagonal matrices $\Phi_X^l := \text{diag}\left(\phi_X^l\right)$, where an element in the diagonal $\phi_X^l := \mathbf{1}\left(\widetilde{x} \geq 0\right)$ is one

if its pre-activation $\widetilde{x}^l := W^l \phi^{l-1} + b^l$ is positive (Roth et al., 2020). Then, the global Lipschitz constant of $g(x)$ is bounded by the spectral norm of following expression (Yoshida & Miyato, 2017; Roth et al., 2020; Huang et al., 2021):

$$L_{\mathrm{glob}} \leq \left\| W^L \cdot \Phi_X^{L-1} \cdot W^{L-1} \cdot \Phi_X^{L-2} \cdots \Phi_X^1 \cdot W^1 \right\|_2 \tag{22}$$

For better comparison with our proposed method, we do not further simplify Eq. (22) like (Yoshida & Miyato, 2017). With the normalization (Ioffe & Szegedy, 2015) in DNN, the mean of final features are always close to zero. Suppose that the mean of deep features is equal to zero, then the proposed FSR denoted as $\ell_{\mathrm{FSR}}$ could be simplified to

$$
\begin{aligned}
\ell_{\mathrm{FSR}} &= \lambda_{\max}\left(F^T F\right) \\
&= \sigma_{\max}\left(W^L \phi^{L-1}\left(W^{L-1}\phi^{L-2}\left(\cdots\cdots\phi^1\left(W^1 x + b_1\right)\cdots\cdots\right)+b^{L-1}\right)+b^L\right)^2 \\
&= \left\|W^L \phi^{L-1}\left(W^{L-1}\phi^{L-2}\left(\cdots\cdots\phi^1\left(W^1 x + b_1\right)\cdots\cdots\right)+b^{L-1}\right)+b^L\right\|_2^2 \\
&= \left\|W^L \cdot \Phi_X^{L-1} \cdot W^{L-1} \cdot \Phi_X^{L-2}\cdots\Phi_X^1 \cdot W^1 \cdot x\right\|_2^2
\end{aligned}
\tag{23}
$$

where the deep features $F = W^L \phi^{L-1}\left(W^{L-1}\phi^{L-2}\left(\cdots\cdots\phi^1\left(W^1 x + b_1\right)\cdots\cdots\right)+b^{L-1}\right)+b^L$, $\lambda_{\max}\left(\cdot\right)$ is the largest eigenvalue of a matrix, and $\sigma_{\max}\left(\cdot\right)$ is the spectral norm (largest singular value) of a matrix.

If we compare Eq. (22) and Eq. (23), FSR has a very similar form with the upper bound of global Lipschitz constant. Since the input data $x$ is a constant, it could be expected that FSR has an effect to decrease the upper bound of the Lipschitz constant. FSR decreases the Lipschitz constant by constraining features rather than directly regularizing weights.

## F.2 CONNECTION WITH ADVERSARIAL EXAMPLES ON MANIFOLD

Our observation is consistent with the hypothesis in (Szegedy et al., 2013; Song et al., 2018) that adversarial examples are low-probability (high-dimensional) "pockets" on the manifold from the perspective of spectral signatures. Our paper reveals that the components along eigenvectors with severely smaller eigenvalues are much rare in standard training (seeing Figure 2). When confronted with attacks, the adversary tries to add more components along with such directions (seeing Figure 4). The eigenvectors with smaller eigenvalues could be regarded as "pockets" on the manifold.

