# OpenReview forum: "Delving into Feature Space: Improving Adversarial Robustness by Feature Spectral Regularization"
_ICLR.cc/2022/Conference — ICLR 2022 Submitted_

### Official Review · Reviewer_wNJf · 2021-10-28

**Correctness:** 2
**Technical Novelty And Significance:** 1
**Empirical Novelty And Significance:** 2
**Recommendation:** 3
**Confidence:** 4

**Main Review:**

The analysis of connections between the sensitivity to attacks and the subspace of the computed embedding is very interesting. I believe that the authors might have struck here a very interesting research direction, which could fuel further research and lead to new insights.

Unfortunately, the present paper does not go beyond the incidental observations. There is no theoretical guarantee or connection which indicates how the dimensionality of the embedding influences the robustness to attacks or how low-variance subspaces can be exploited by attack methods. The authors do provide a theoretical analysis for robust linear regression models, but the connection/applicability to DNNs is unclear and I also come to the opposite conclusion from the authors, regarding the eigenvalues of the most robust regression model (see below).

The experiments show that the proposed regularizer can improve the robustness by a few percent, but we do not observe that a more homogeneous distribution of eigenvalues really prevents adversarial attacks from happening. In this respect, I doubt that the proposed method would be used in practice. Employing the regularizer requires tuning its weight, and even when the authors could show that for Cifar-10, -100 and SVHN a weight around 0.01 is suitable, this might look very different for other datasets. Hence, a theoretical guarantee/connection from the embedding PCA to robustness would be needed in order to estimate in which cases the proposed approach can help and to what extend.

### Theorem 1
Theorem 1 states that the design matrix $Z$ of a linear regression model yields the most robust predictions if all eigenvalues of $Z$ are equal. The robustness is measured by $\lVert Z\theta_a -Z\theta_0\rVert^2$, where $\theta_a = (\Sigma+\lambda I )^{-1}\Sigma\theta_0$. Using the SVD of $Z=UDV^\top$, we can rewrite
$$
\begin{align*}
\lVert Z\theta_a -Z\theta_0\rVert^2 &= \lVert \mathrm{diag}\left(\frac{n\lambda\sigma_i}{\sigma_i^2+n\lambda}\right)V^\top \theta_0\rVert^2\\
&= \sum_i \left( \frac{n\lambda\sigma_i}{\sigma_i^2+n\lambda} V_{\cdot i}^\top \theta_0\right)^2\\
&= \sum_i  \frac{n^2\lambda^2\sigma_i^2}{(\sigma_i^2+n\lambda)^2} (V_{\cdot i}^\top \theta_0)^2
\end{align*}
$$
The last term attains its minimizer $Z$ s.t. the constraint $\lVert Z\rVert=s_0$ if $\sigma_1=s_0$, $V_{\cdot 1}$ orthogonal to $\theta_0$ and the rest of the singular values being zero. In your proof, you fix $y=z^\top \theta_0$, but since you optimize over $Z$ and since $y$ is a function of $z$ I don't see how you can treat $y$ like a constant.

### Minor issues:
* Use maybe grammarly or another tool to capture the most obvious language errors
* The space in which PCA is applied and whose eigenvalues are inspected in this work remains largely unclear until Section 3.
* Is there any connection of the PCA indicators to known robustness influencers, like the Lipschitz constant of the DNN function for example?
* Can we infer something about the manifold hypothesis (natural examples are on a manifold in the embedded space and adversarial examples are outside of that manifold) from the PCA of the embedding? How can your observation be placed in the scope of the related work on adversarial examples on manifolds?
* How can we transfer the information about robust linear regression models to DNNs?
* Why is FSR not evaluated without an additional defense method?





**Summary Of The Paper:**

The authors propose a new defense against adversarial attacks by means of a spectral regularization. The defense is based on an inspection of the embedding of training/test data as returned by the penultimate layer of a neural network. The observation is that the relevant subspace of embedded natural data points (covering e.g. 90% of the variance) is quite low-dimensional, where the use of adversarial training increases the dimensionality of this subspace. In addition, the relevant subspace of embedded adversarial examples is in the provided experiments typically a bit higher dimensional than the subspace of natural examples. Based on these observations, the authors propose a regularizing term, which penalizes the variance in the direction of the first principal component (the largest eigenvalue of the feature covariance matrix).

Experiments are conducted on Cifar-10, -100 and SVHN. The proposed regularizer is applied on top of other defense methods, which typically increases the accuracy under attacks by up to 2%.

**Summary Of The Review:**

The authors make an interesting observation about adversarial robustness of DNN classifiers based on a PCA of the embedded data points. However, the observations are not backed up by theoretical insights and the empirical evidence is not very conclusive.

---

> ### Author Response · Authors · 2021-11-20
> **Response to Reviewer wNJf (2/2)**
>
> **Question 3:  The connection between our method to known robustness influencers, like Lipschitz constant**
> Thanks for pointing out this issue. We have delved into this perspective during the rebuttal phase and a concise explanation based on the Lipschitz constant is provided here. **The detailed analysis has been added in Appendix F.1.**
> For a feature extractor $g(x):\mathbb{R}^D\rightarrow \mathbb{R}^d$ composed of $L$ layers, the global Lipschitz constant $L_{\mathrm{glob}}$ of $g(x)$ is bounded by the spectral norm of weights [Yoshida &Miyato, 2017; Roth et al., 2020; Huang et al., 2021]:
> $$
> L_{\mathrm{glob}}\le \|\| W^L\cdot \Phi _{X}^{L-1}\cdot W^{L-1}\cdot \Phi _{X}^{L-2}\cdots \Phi _{X}^{1}\cdot W^1 \|\| _2,$$
> where the norm is the spectral norm (largest singular value) of a marix, $W^i (i=1,\cdots,L)$ is the weight of layer $i$, and $\Phi _{X}^{l}$ is a function defined by [Roth et al., 2020] whose details are included in Appendix F.1.
> With the batch normalization in DNN, it is reasonable to assume that the means of final features are close to zero. Suppose that the mean of deep features is equal to zero, FSR has a similar form as the upper bound of Lipschitz constant:
> $$
> \ell _{\mathrm{FSR}}=\|\| W^L\cdot \Phi _{X}^{L-1}\cdot W^{L-1}\cdot \Phi _{X}^{L-2}\cdots \Phi _{X}^{1}\cdot W^1\cdot x \|\| _{2}^{2}.
> $$
> The proposed method has a similar formula with the upper bound of the Lipschitz constant. Since the input $x$ is a constant, it could be expected that FSR tries to decrease the upper bound of the Lipschitz constant. FSR decreases the Lipschitz constant by constraining features rather than directly regularizing weights.
>
> Reference:
> [Yoshida &Miyato, 2017] Spectral norm regularization for improving the generalizability of deep learning, arxiv, 2017.
> [Roth et al., 2020] Adversarial training is a form of data-dependent operator norm regularization. NeurIPS 2020.
> [Huang et al., 2021] Training certifiably robust neural networks with efficient local lipschitz bounds. NeurIPS 2021.
>
>
> **Question 4: Connection with adversarial examples on manifolds**
> Our observation is consistent with the hypothesis proposed by [Szegedy et al., 2013; Song et al., 2018] that adversarial examples are low-probability (high-dimensional) "pockets" on the manifold, from the perspective of spectral signatures. Our paper reveals that the components along eigenvectors with severely smaller eigenvalues are much rare in standard training (seeing Figure 2). When confronted with attacks, the adversary tries to add more components along such directions (seeing Figure 4). Therefore, the eigenvectors with smaller eigenvalues could be regarded as "pockets" on the manifold. We will move the discussion to the main body of the final paper.
>
> Reference:
> [Szegedy et al., 2013] Intriguing properties of neural networks. ICLR 2014.
> [Song et al., 2018] Pixeldefend: Leveraging generative models to understand and defend against adversarial examples. ICLR 2018.
>
> **Question 5: Effectiveness without extra defense method**
> We have shown that using the proposed method without AT could achieve a mild improvement in adversarial robustness and confidence calibration. The detailed results are provided in Appendix D.
>
> **Other minor issues**
>  - **About language errors**
>  Thanks for pointing out this, we will check the paper carefully and thoroughly.
>  - **Whose eigenvalues are inspected in the work remains unclear until Section 3.**
>  We will revise the paper and move the descriptions of the details about how to use PCA to an earlier place in the paper.

---

> > ### Comment · Reviewer_wNJf · 2021-11-23
> > **Effectiveness without extra defense method**
> >
> > Dear authors,
> >
> > Thank you for implementing my suggestions. I found in particular the experiments showing that your method is able to provide confidence calibration very interesting and relevant for future research directions. I am still wondering if you have an explanation/idea/intuition about the experimental outcome that the effect of your method on robustness is not so strong. Don't these experimental results contradict your hypothesis that adversarial examples exploit the low-variance directions in the embedded feature space? Or how do you interpret these results?

---

> > > ### Author Response · Authors · 2021-11-24
> > > **Re: Effectiveness without extra defense method**
> > >
> > > Thanks a lot for your time and valuable comments. These results do not contradict our hypothesis that adversarial examples exploit the low-variance directions in the embedded feature space. The reasons are listed as follows:
> > >
> > >  - **Simply using FSR without adversarial training (AT) could attain a significant improvement on robustness.** As shown in Table 8, standard training with FSR boosts the performance on defending against FGSM (**+4.58%** when $\epsilon=2/255$, from 54.61% to 59.19%). The improvement on FGSM could be regarded as evidence that the robustness of the model is improved, as used in recent work [Li et al., 2021]. It is reasonable to expect that simply using FSR based on standard training could not achieve the same performance as AT (seeing more in the explanation below). For example, the work of [Hendrycks et al., 2019] also offered an analysis of this. As illustrated in [Hendrycks et al., 2019], simply using pre-training would not attain the same performance as AT, but when pre-training is combined with AT, the performance of AT is improved. The main purpose of FSR is to assist AT, so the main experiments in the paper are based on AT.
> > >
> > >  - **Adversarial training is essential in defending against strong attacks.** When confronted with very strong attacks, nearly all defenses fail [Athalye et al., 2018] except AT. However, training a robust model is much harder than the standard training, e.g., AT requires more data and much more complicated networks [Schmidt et al. 2018; Xie and Yuille 2020]. Since using AT is not enough for a robust model, there are many approaches to improve the robustness of the model based on AT. The main approaches include adding more data and regularization. Our work aims to explore the property which is not attracted enough attention by AT and propose a method to assist AT. Since understanding the practice of AT is a very hard task [Robey et al., 2021], there may be other properties of AT that could not be observed from the perspective of spectral signatures.  Our work could not completely solve the problem of adversarial robustness, but it has provided new and unique insights to analyse AT.
> > >
> > >  - **The experimental results and theoretical analysis in the paper are consistent.** Our paper tries to analyse the adversarial examples and adversarial robustness of models from the perspective of spectral signatures. In this paper, we find that the dominance of the top eigenvalues in feature space is harmful to adversarial robustness, and propose FSR to alleviate the phenomenon in the distribution of eigenvalues. The results show that FSR indeed alleviates the dominance of the top eigenvalues (e.g., increasing the estimated intrinsic dimension from 22 to 62 in Figure 2(a)), and the adversarial robustness of models is also improved (Table 1-3, 9-11).
> > >
> > > Overall, the results of the experiments verify the hypothesis proposed in the paper.
> > >
> > > Thanks again for pointing out this issue, we will add more discussion about the experimental results in the paper. If parts of the response are still unclear, please let us know and we are happy to follow up. We hope you find the response satisfactory and increase the score accordingly.
> > >
> > > Reference:
> > > [Li et al., 2021] Shape-texture debiased neural network training. ICLR 2021.
> > > [Athalye et al., 2018] Obfuscated gradients give a false sense of security: Circumventing defenses to adversarial examples. ICML 2018.
> > > [Hendrycks et al., 2019] Using pre-training can improve model robustness and uncertainty. ICML 2019.
> > > [Schmidt et al. 2018] Adversarially Robust Generalization Requires More Data. NeurIPS 2018.
> > > [Xie and Yuille 2020] Intriguing Properties of Adversarial Training at Scale. ICLR 2020.
> > > [Robey et al., 2021] Adversarial Robustness with Semi-Infinite Constrained Learning. NeurIPS 2021.

---

> ### Author Response · Authors · 2021-11-20
> **Response to Reviewer wNJf (1/2)**
>
> Thank you for the valuable feedback. Your approval that we have struck a very interesting research direction will motivate us to further conduct research between spectral signatures and robustness. Next, we respond to each of your questions and concerns one by one and hope you find the response satisfactory and increase the score accordingly. If parts of them are still unclear, please let us know and we are happy to follow up. A revision of our paper has also been uploaded.
>
> **Question 1:  Explanation about Theorem 1 in Section 4.2**
> We would like to clarify that the reviewer may have some misunderstandings about Theorem 1 beacuase of our unclear notations. The output of linear regression is modified to be $\hat{y}$ (rather than $y$), so the linear model is $\hat{y}=\left< z,\theta \right>$. $y=[y_1, y_2, \cdots, y_n]^T$ is a column vector composed of real-valued output $y_i\left( i=1,\cdots ,n \right)$.  We have changed these notations in the paper. $\theta _0$ is obtained by minimizing mean square error rather than a constant i.e., $\theta _0=\left( Z^TZ \right) ^{-1}Z^Ty$.
> Following the derivation in [Hastie et al., 2009], the results provided by the reviewer could be simplified to
> $$
> \|\| Z\theta_a-Z\theta _0 \|\|_2 = \|\| \sum_i {u_i }\frac{n\lambda}{\sigma _{i}^{2}+n\lambda} u_i^{T} y\|\|_2 \le \|\| \sum_i {u_i }\frac{n\lambda}{\sigma _{i}^{2}+n\lambda} u_i^{T}\|\|_2 \cdot \|\| y \|\|_2 =\frac{n\lambda}{\sigma _{d}^{2}+n\lambda}\cdot \left\|\| y \right\|\| _2,
> $$
> Consider that $y$ could be a constant of arbitrary value, we need to minimize $\frac{n\lambda}{\sigma _{d}^{2}+n\lambda}$, i.e., $\sigma_d$ should be as large as possible. Since it is the minimum singular value of $Z$, the adversarial risk induced by $Z$ is minimal when all singular values are equal. Detailed proof could be found in Appendix C. The theoretical analysis uncovers the connection between the embedding PCA to adversarial robustness.
> We believe such misunderstanding is partially due to the presentation in Section 4.2, which is not clear enough. We have changed some notations in this part, and hope our explanation can help the reviewer reevaluate our work.
>
> Reference:
> [Hastie et al., 2009] The Elements of Statistical Learning: Data Mining, Inference, and Prediction. *Springer*, 2009.
>
> **Question 2: The paper does not go beyond the identical observation. The authors do provide theoretical analysis for robust linear regression models, but the connection to DNNs is unclear.**
> We would like to clear up the main concern about our analysis. It has been shown that the phenomenon that adversary tends to add more components along the eigenvectors with smaller eigenvalues could be found in the various dataset (CIFAR-10, CIFAR-100, and SVHN) in the paper, and we have provided a theoretical guarantee of FSR, i.e., Theorem 1 in Section 4.2. Therefore, the conclusion in the paper does not come from the identical observation.
>
> Furthermore, the theoretical analysis of linear case has a close connection with DNN. Three main reasons are elaborated:
>  - **Theoretical analysis in the linear case is meaningful and widely used.** Many theoretical works have adopted the linear case for analytical solutions since simple settings can manifest as special cases of more complex settings. For example, [Schmidt et al., 2018; Tsipras et al., 2019; Xu et al., 2021] used a linear model to theoretically analyse sample complexity of robust generalization, the tradeoff between robustness and accuracy, and unfairness problem in AT, respectively.
>  - **Current analysis is nontrivial.** We would like to note that our analysis is novel, and the result based on linear case has provided some new insights. Extending the analysis to DNNs is hard and interesting, especially for adversarial learning, so it is better suited as a future direction. Our theoretical analysis is also acknowledged by Reviewer kop5 and Reviewer gwLs.
>  - **The analysis could be understood from the perspective of representation learning.** We just replace the linear classifier and cross-entropy in DNN with linear regression and mean square loss. The features fed to linear regression could be regarded as deep features or representations gotten by DNN, and a similar method has been used in [Yu et al., 2020]. Theorem 1 points out that when the singular values of representation are equal, the adversarial risk induced by representation is minimal, i.e., the features extracted by DNN should have equal singular values.
>
> Reference:
> [Schmidt et al., 2018] Adversarially robust generalization requires more data. NeurIPS 2018.
> [Tsipras et al., 2019] Robustness may be at odds with accuracy. ICLR 2019.
> [Xu et al., 2021] To be Robust or to be Fair: Towards Fairness in Adversarial Training. ICML 2021.
> [Yu et al., 2020] Learning diverse and discriminative representations via the principle of maximal coding rate reduction. NeurIPS 2020.

---

> > ### Comment · Reviewer_wNJf · 2021-11-23
> > **Theorem 1**
> >
> > Dear authors,
> >
> > Thanks for your explanations and revisions. I just still don't understand the addressed point in the proof of Thm 1. In the simplifications, you assume that $Z\theta_0 = Z(Z^\top Z)^{-1}Z \theta_0 = Y$, right? This equation does however not always hold (otherwise all regression models would always perfectly fit the training data), which is why I assumed that you mean with Y the predictions returned by the model. Are there some assumptions I missed such that the equation above holds?
> >
> > Thank you for your answers.

---

> > > ### Author Response · Authors · 2021-11-24
> > > **Response to Theorem 1**
> > >
> > > Thank you very much for the response. We would like to clear up the main concern about our theoretical analysis. In the simplifications, we do not assume that $Z\theta_0=y$. In the following part, a detailed derivation about the connection between $Z\theta_0$ and $y$ is provided.
> > > $\theta_0$ is obtained by minimizing mean square error, i.e., $\theta_0=\left( Z^TZ \right) ^{-1}Z^Ty$. $y=\left[ y_1,y_2,\cdots ,y_n \right]$ is a column vector composed of real-valued output $y_i\left( i=1,\cdots ,n \right)$. Suppose $d<n$, then $Z=UDV^T\in\mathbb{R}^{n\times d}$ after applying SVD, where $U=\left[ u_1,\cdots ,u_n \right] \in \mathbb{R}^{n\times n}$ and $V\in\mathbb{R}^{d\times d}$ are orthogonal matrices, and $D=\left[ \begin{array}{c}
> > > 	D_0\\\\
> > > 	0\\
> > > \end{array} \right] \in \mathbb{R}^{n\times d}$. $D_0=\mathrm{diag}\left( \sigma _1,\cdots ,\sigma _d \right)$, and we suppose $\sigma _i>0\left( i=1,\cdots ,d \right)$, i.e., $D_0$ is a nonsingular matrix. Then
> > >
> > > $$
> > > \begin{aligned}
> > > Z\theta_0&=Z\left( Z^TZ \right) ^{-1}Z^Ty \\\\
> > > &=\left( UDV^T \right) \left( VD^TU^TUDV^T \right) ^{-1}\left( VD^TU^T \right) y \\\\
> > > &=UDV^T\left( VD_{0}^{2}V^T \right) ^{-1}VD^TU^Ty \\\\
> > > &=U\left[ \begin{array}{c}
> > > 	D_0\\\\
> > > 	0\\
> > > \end{array} \right] V^TVD_{0}^{-2}V^TV\left[ \begin{matrix}
> > > 	D_0&		0\\
> > > \end{matrix} \right] U^Ty \\\\
> > > &=U\left[ \begin{array}{c}
> > > 	D_0\\\\
> > > 	0\\
> > > \end{array} \right] D_{0}^{-2}\left[ \begin{matrix}
> > > 	D_0&		0\\
> > > \end{matrix} \right] U^Ty \\\\
> > > &=U\left[ \begin{matrix}
> > > 	I^{d\times d}&		0\\\\
> > > 	0&		0^{\left( n-d \right) \times \left( n-d \right)}\\
> > > \end{matrix} \right] U^Ty \\\\
> > > &=\left[ u_1,\cdots ,u_n \right] \left[ \begin{matrix}
> > > 	I^{d\times d}&		0\\\\
> > > 	0&		0^{\left( n-d \right) \times \left( n-d \right)}\\
> > > \end{matrix} \right] \left[ u_1,\cdots ,u_n \right] ^Ty \\\\
> > > &=\left[ u_1,\cdots ,u_d,0,\cdots ,0 \right] \left[ u_1,\cdots ,u_n \right] ^Ty \\\\
> > > &=\sum_{i=1}^d{u_iu_{i}^{T}y}.
> > > \end{aligned}
> > > $$
> > > The detailed derivation could also be found in [Hastie et al., 2009].
> > > **It should be noticed that $\sum_{i=1}^n{u_iu_{i}^{T}y}=UU^Ty=y$, while $Z\theta_0=\sum_{i=1}^d{u_iu_{i}^{T}y}\ne y$. The equation $Z\theta_0=y$ holds if $d=n$, while it is contradictory with our assumption that $d<n$. The estimated output $Z\theta_0$ is the orthogonal projection onto the subspace spanned by the orthogonal basis $\left\\{ u_1,\cdots ,u_d \right\\}$ rather than $\left\\{ u_1,\cdots ,u_n \right\\}$** [Hastie et al., 2009]. In addition, we could apply a similar deviration to simplify $Z\theta_a$, and then $Z\theta_a=\sum_{i=1}^d{u_i\frac{\sigma_{i}^{2}}{\sigma_{i}^{2}+n\lambda}}u_{i}^{T}y$.
> > >
> > > We believe such misunderstanding is partially due to the lack of details in the proof, and we will add more details about the theoretical analysis in the final paper. If parts of them are still unclear, please let us know and we are happy to follow up.
> > >
> > > Reference:
> > > [Hastie et al., 2009] The Elements of Statistical Learning: Data Mining, Inference, and Prediction. *Springer*, 2009.

---

> > > > ### Comment · Reviewer_wNJf · 2021-11-24
> > > > **Theorem 1**
> > > >
> > > > Ok, so you don't assume that $Z\theta_0=Y$, but how do you then get the result of Thm 1? Let me maybe state hoow I thought your proof goes, then you can tell me where I'm wrong:
> > > > $\theta_{adv} = (\frac{1}{n}ZZ^\top +\lambda I )^{-1}\frac{1}{n}Z^\top Z\theta_0$ , and the SVD yields $Z=UDV^\top$, where $D=\mathrm{diag}(\sigma_i)$ then we have:
> > > > $\begin{aligned}
> > > > Z\theta_{adv} &= UDV^\top V \mathrm{diag}\left(\frac{n}{\sigma_1^2+\lambda n}\right)V^\top \frac{1}{n} VD^\top U^\top Z\theta_0 \\\\
> > > > &=  U \mathrm{diag}\left(\frac{\sigma_i^2}{\sigma_1^2+\lambda n}\right) U^\top Z\theta_0
> > > > \end{aligned}$
> > > > Then we compute
> > > > $\begin{align}
> > > > \lVert Z\theta_{adv}-Z\theta_0\rVert & = \lVert(U \mathrm{diag}\left(\frac{\sigma_i^2}{\sigma_1^2+\lambda n}\right) U^\top - I)Z\theta_0\rVert\\\\\
> > > > &= \lVert U\mathrm{diag}\left(-\frac{n\lambda}{\sigma_i^2+n\lambda}\right)UZ\theta_0\rVert\\\\
> > > > &= \lVert \mathrm{diag}\left(\frac{n\lambda}{\sigma_i^2+n\lambda}\right)UZ\theta_0\rVert
> > > > \end{align}$
> > > > where the last equation follows from the orthogonal invariance of the norm. Now, we get exactly the result you stated when we assume that $Z\theta_0=Y$, because then we have
> > > > $$\lVert \mathrm{diag}\left(\frac{n\lambda}{\sigma_i^2+n\lambda}\right)UZ\theta_0\rVert \leq \lVert \mathrm{diag}(\frac{n\lambda}{\sigma_i^2+n\lambda})\rVert \lVert Y \rVert$$
> > > > In your previous reply, you showed that $Z\theta_0 = U_{\cdot :d}U_{\cdot :d}^\top Y$, so $U^\top Z\theta_0 = U^\top U_{\cdot :d}U_{\cdot :d}^\top Y$ which is also not equal to $Y$, so how do you do that last step then?

---

> > > > > ### Author Response · Authors · 2021-11-24
> > > > > **Re: Theorem 1**
> > > > >
> > > > > Thanks very much for your prompt reply.  We would like to clear up the concern about the proof.
> > > > >
> > > > > Where the reviewer made a mistake is that the final inequality, i.e.,
> > > > > $$\|\| \mathrm{diag}\left( \frac{n\lambda}{\sigma _{i}^{2}+n\lambda} \right) U^TZ\theta _0 \|\| \le \|\| \mathrm{diag}\left( \frac{n\lambda}{\sigma _{i}^{2}+n\lambda} \right) \|\| \~ \|\| Y \|\|,
> > > > > $$
> > > > >
> > > > > **does not require** $Z\theta_0=y$. The detailed proof of the inequality is provided as follows.
> > > > > $$
> > > > > \begin{aligned}
> > > > > \|\| \mathrm{diag}\left( \frac{n\lambda}{\sigma _{i}^{2}+n\lambda} \right) U^TZ\theta _0 \|\| &\le \|\| \mathrm{diag}\left( \frac{n\lambda}{\sigma _{i}^{2}+n\lambda} \right) U^T \|\| \~ |\| Z\theta _0 \|\|
> > > > > \\\\
> > > > > &= \|\| \mathrm{diag}\left( \frac{n\lambda}{\sigma _{i}^{2}+n\lambda} \right) \|\| \~ \|\| Z\theta _0\|\|
> > > > > \end{aligned}
> > > > > $$
> > > > > **Remark:** The matrix norm used is the spectral norm, and the vector norm used is the Euclidian norm. The spectral norm is the matrix norm induced by the Euclidian norm.
> > > > >
> > > > > ***The first step above holds*** because of the property of matrix norm induced by vector norm, i.e., $\|\| Ax \| \| _2\le \|\| A \|\| _2 \~ \|\| x \|\| _2$, where $A$ is a $m\times n$ matrix and $x$ is a $n$-dimensional vector. ***The second step*** holds because of the orthogonal invariance of the norm.
> > > > >
> > > > > Then, since $Z\theta_0=U _{\cdot :d}U _{\cdot :d}^{T}Y$, we could further simplify the expression.
> > > > > $$
> > > > > \begin{aligned}
> > > > > \|\| \mathrm{diag}\left( \frac{n\lambda}{\sigma _{i}^{2}+n\lambda} \right) \|\| \~ \|\| Z\theta _0\|\|  &= \|\| \mathrm{diag}\left( \frac{n\lambda}{\sigma _{i}^{2}+n\lambda} \right) \|\| \~ \|\| U _{\cdot :d}U _{\cdot :d}^{T}Y \|\|
> > > > > \\\\
> > > > > &\le \|\| \mathrm{diag}\left( \frac{n\lambda}{\sigma _{i}^{2}+n\lambda} \right) \|\| \~ \|\| U _{\cdot :d}U _{\cdot :d}^{T} \|\| \~\|\| Y \|\|
> > > > > \\\\
> > > > > &= \|\| \mathrm{diag}\left( \frac{n\lambda}{\sigma _{i}^{2}+n\lambda} \right)\|\| \~ \|\| Y \|\|
> > > > > \end{aligned}
> > > > > $$
> > > > >
> > > > > ***The second step above holds*** because of the property of matrix norm induced by vector norm, i.e., $\|\| U_{\cdot :d}U_{\cdot :d}^{T}Y \|\| \le \|\| U _{\cdot :d}U _{\cdot :d}^{T} \|\| \~ \|\| Y \|\|$. ***The third step*** holds because *all non-zero eigenvalues of $U _{\cdot :d}U _{\cdot :d}^{T}$ are equal to $1$. As a real symmetric matrix, the spectral norm of $U _{\cdot :d}U _{\cdot :d}^{T}$ is equal to the absolute value of maximum eigenvalue. Therefore, $\|\| U _{\cdot :d}U _{\cdot :d}^{T} \|\| =1$*. **Explanations about the eigenvalues of $U _{\cdot :d}U _{\cdot :d}^{T}$ are provided as follows.**
> > > > >
> > > > > Next, we talk about how to attain the non-zero eigenvalues of the matrix $U _{\cdot :d}U _{\cdot :d}^{T}$. It should be noticed that $U _{\cdot :d}=\left[ u_1,\cdots ,u_d \right]$. In addition, $u _{i}^{T}u _j=1$ if $i=j$, and $u _{i}^{T}u _j=0$ if $i\ne j$. Then the following expression holds:
> > > > > $$
> > > > > \left( U _{\cdot :d}U _{\cdot :d}^{T} \right) u_j = \left( \sum _{i=1}^d{u _iu _{i}^{T}} \right) u_j=1u_j \~ (j=1,\cdots, d).
> > > > > $$
> > > > > Therefore, according to the definition of eigenvalues, the $d$ non-zero eigenvalues of $U _{\cdot :d}U _{\cdot :d}^{T}$ are equal to 1, and its corresponding eigenvectors are $\\{ cu_1,\cdots ,cu_d \\}$, where $c$ is a non-zero constant. Since the rank of $U _{\cdot :d}U _{\cdot :d}^{T}$ is equal to $d$, the other $n-d$ eigenvalues of $U _{\cdot :d}U _{\cdot :d}^{T}$ all equal to 0. As a result, the spectral norm of $U _{\cdot :d}U _{\cdot :d}^{T}$ equals to 1, i.e., $\|\| U _{\cdot :d}U _{\cdot :d}^{T} \|\| =1$.
> > > > >
> > > > > **According to the above analysis, the final inequality does not require $Z\theta_0=y$.**
> > > > >
> > > > > Overall, we believe the misunderstanding from the reviewer is *due to the lack of some key information in the proof*. Thanks again for pointing out this issue, we will provide more detailed proof in the final paper which can help the reader better understand our theoretical analysis.
> > > > >
> > > > > If parts of them are still unclear, please let us know and we are happy to follow up.

---

> > > > > > ### Comment · Reviewer_wNJf · 2021-11-26
> > > > > > **Theorem 1**
> > > > > >
> > > > > > Thank you, I get these inequalities, and I also see that what I wrote in my original review was not correct, since I treated $\theta_0$ like a constant. Nevertheless, there is still one issue. If you want to find a minimizer of a function, you can not minimize an upper bound of that function and then assume that the minimizer of the upper bound and the function is the same. Because if you don't apply any inequalities, then I come to the following term (using the transformations of two replies ago and what you derived for $Z\theta_0$):
> > > > > >
> > > > > > $$\begin{align}
> > > > > > \lVert Z\theta_{adv} - Z\theta_0\rVert^2 & = \lVert \mathrm{diag}\left(\frac{n\lambda}{\sigma_i^2+n\lambda}\right)U^\top Z\theta_0\rVert^2\\\\
> > > > > >  & =  \lVert \mathrm{diag}\left(\frac{n\lambda}{\sigma_i^2+n\lambda}\right)U^\top U \begin{pmatrix} I_d & 0\\\\\ 0&0\end{pmatrix}U^\top Y\rVert^2\\\\
> > > > > > & = \sum_{i=1}^d \left(\frac{n\lambda}{\sigma_i^2+n\lambda}\right)^2 (U_{\cdot i}^\top Y)^2
> > > > > > \end{align}$$
> > > > > > I don't think that choosing all singular values the same is the minimum of this function. The term can be minimized by choosing the singular values large when the inner product $U_{\cdot i}^\top Y$ is high and vice versa.
> > > > > > Since this is your only theoretic result, I think it should really stand oon solid ground. Maybe this is fixable and I can update my score.

---

> > > > > > > ### Author Response · Authors · 2021-11-27
> > > > > > > **Re: Theorem 1**
> > > > > > >
> > > > > > > Thanks for your valuable comments.  It is great to hear that the concerns about these inequalities have been addressed. We would like to explain the theoretical results as follows.
> > > > > > >
> > > > > > >  - **It is hard to get an analytical solution based on the simplified formula.** The formula is expressed as follows. It is worth noticing that there is a constraint on the sum of singular values to control the scale of features as used in [Yu et al., 2020].
> > > > > > >  $$
> > > > > > > \|\| Z\theta_{adv}-Z\theta_0 \|\|^2=\sum_{i=1}^d{\left( \frac{n\lambda}{\sigma_{i}^{2}+n\lambda} \right) ^2}\left( U _{\cdot :i}^{T}Y \right) ^2 ,\quad s.t. \sum _{i=1}^d {\sigma _{i}^{2}=s_0}
> > > > > > >  $$
> > > > > > >
> > > > > > > As illustrated by the reviewer, the term can be minimized by choosing the singular values large when the inner product $U_{\cdot :i}^{T}Y$ is high and vice versa. However, it is very hard to directly solve this problem, because there is a coupling relationship between between $\left\\{ \sigma _1,\cdots ,\sigma _d \right\\}$ and $U _{\cdot :i}$. After applying SVD, $Z=UDV^T$.  $\left\\{ \sigma _1,\cdots ,\sigma _d \right\\}$ and  $U _{\cdot :i}$ have a close connection with $D$ and $U$, respectively. **If we change the value of $\left\\{ \sigma _1,\cdots ,\sigma _d \right\\}$, $U _{\cdot :i}^TY$ may also be changed.** Since we keep regarding $Y$ as a fixed value, finding the minimum of $\|\| Z\theta _{adv}-Z\theta _0 \|\|^2$ is very challenging.
> > > > > > >
> > > > > > >  - **The solution in the paper is meaningful.** Since we could not get an analytical solution about the original problem, applying a relaxation to *eliminate the influence of $Y$ on the result* is reasonable. Based on the concept of worst-case optimization, we try to minimize the maximum risk induced by a special $Y$. The following formula is an upper bound and the equal sign is easy to set up. Therefore, an analytical solution can be obtained.
> > > > > > > $$
> > > > > > > \|\| Z\theta_{adv}-Z\theta_0 \|\| ^2\le \sum_{i=1}^d {\left( \frac{n\lambda}{\sigma_{i}^{2}+n\lambda} \right)}^2 \left( \|\| U _{\cdot :i}^{T} \|\| \~ \|\| Y \|\| \right) ^2 = \sum _{i=1}^d {\left( \frac{n\lambda}{\sigma _{i}^{2}+n\lambda} \right)}^2\|\| Y \|\| ^2
> > > > > > > $$
> > > > > > >
> > > > > > > The solution is in fact the minimum of the upper bound of the original problem. We will modify the expression of Theorem 1 in the paper to ***"The upper bound of $\mathcal{R}_{res}\left( \sigma _1,\cdots ,\sigma _d \right)$ is minimum when all the singular values of $Z$ are equal."*** The theorem is also meaningful. On the one hand, we may meet such a special $Y$ that contributes to the worst-case. On the other hand, worst-case optimization is a common analysis method in adversarial settings. In addition, we have verified it by comprehensive experiments that the proposed FSR could contribute to an improvement on adversarial robustness.
> > > > > > >  - **We will provide more explanations about our method from other perspectives.** Thanks to the reviewer to help us find more theoretical explanations. We have provided the explanations from the perspective of Lipschitz constant and manifold. These findings help us understand more about the results in the paper. We will add these parts to the main body of the paper in Section 4.2 to provide a more comprehensive theoretical explanation. These theoretical explanations can help readers better understand our paper.
> > > > > > >
> > > > > > > During the discussion with the reviewer, we have a deeper understanding of the theoretical explanation of this work. The reviewer has pointed some issues about the inaccurate statements in theoretical analysis, and guided us to find more explanation/intuition about the experimental outcome and theoretical analysis. We hope you find the response satisfactory and increase the score accordingly.
> > > > > > >
> > > > > > > Thanks again for your dedication to our work during the discussion stage.
> > > > > > >
> > > > > > > Reference:
> > > > > > >
> > > > > > > [Yu et al., 2020] Learning diverse and discriminative representations via the principle of maximal coding rate reduction. NeurIPS 2020.

---

> > > > > > > > ### Comment · Reviewer_wNJf · 2021-11-28
> > > > > > > > **Theorem 1**
> > > > > > > >
> > > > > > > > But can't we just choose $U_{\cdot n}=Y$ and then complete the orthogonal matrix $U$, choosing $\sigma_n=0$? Then we have $U_{\cdot i}^\top Y=0$ for $i\leq d$ and the minimum is attained at value zero and the remaining distribution of the singular values doesn't matter at all for the most robust regression model?
> > > > > > > >
> > > > > > > > In general, the upper bound of a function can have very different minimizers than the function itself.

---

> > > > > > > > > ### Author Response · Authors · 2021-11-30
> > > > > > > > > **Re: Theorem 1**
> > > > > > > > >
> > > > > > > > > Thanks very much for your prompt reply. We would like to address the latest concerns in your reply.
> > > > > > > > >
> > > > > > > > >  - **The special case described by the reviewer is vague and impractical.** The reviewer chooses $U_{\cdot n}=Y$, but this equation is confusing. $U_{\cdot n}$ is matrix while $Y$ is a column vector. The equation $U_{\cdot n}=Y$ could never hold. If we neglect the mistake in the equation, the reviewer then proposes a special case that $U_{\cdot i}^{T}Y=0$. What does this mean in a linear regression model? In this case, the output  of linear regression model will collapse to 0, i.e., $Z\theta_0=U_{\cdot i}U_{\cdot i}^{T}Y=U_{\cdot i}\left( U_{\cdot i}^{T}Y \right) =0$. ***As far as we know, it is meaningless that the output of a linear regression collapses to 0 in practice.*** ***However, this special case is illustrated by the reviewer as a representative of "the most robust regression model"  to negate the meaning of our conclusion.*** We do not agree with the reviewer at this point. This special case is impractical, and it is less likely to say that this belongs to "the most robust regression model".
> > > > > > > > >  - **It is acceptable to derive an upper bound for better analysis.** We derive an upper bound of the adversarial risk defined by us in this paper and try to analyse the minimum of the bound. Since the original problem is untraceable, deriving an upper bound is a common method for better analysis. Strictly speaking, the solution of the upper bound may be different from the original problem under some special cases, but it is acceptable. There are many insightful works that try to find an upper bound of risk to solve the problem of robustness. For example, the work of [Wong & Kolter, 2018] used convex relaxation to attain provable robustness. They formed a relaxation of the activation function ReLU and obtained an upper bound of the inner optimization problem in adversarial training. The work of [Zhang et al., 2021] defined the adversarial risk and then proved that mixup could approximately minimize the upper bound of the adversarial risk, thus explaining how mixup could improve robustness. For an untraceable problem, finding the bound of the original problem is feasible.
> > > > > > > > >
> > > > > > > > > Thanks again for your dedication to our work during the rebuttal process. We have tried our best to respond to all the questions, from explaining the proof of our theoretical result to adding more theoretical results as suggested by the reviewer. We hope you can re-evaluate our work.
> > > > > > > > >
> > > > > > > > > Reference:
> > > > > > > > > [Wong & Kolter, 2018] Provable defenses against adversarial examples via the convex outer adversarial polytope. ICML 2018.
> > > > > > > > > [Zhang et al., 2021] How does mixup help with robustness and generalization?. ICLR 2021.

---

> > > > > > > > > > ### Comment · Reviewer_wNJf · 2021-11-30
> > > > > > > > > > **Theorem 1**
> > > > > > > > > >
> > > > > > > > > > Dear authors,
> > > > > > > > > >
> > > > > > > > > > I think we have reached the end of feasibility of discussing a proof over openreview. But let me just set my argument right, because you misunderstood what I wrote.
> > > > > > > > > > First of all, with $U_{\cdot n}$ I denote the $n$th column of $U$, so this should have the same dimension like $Y$ and hence I do not see why the solution I suggested would not be correct.
> > > > > > > > > > Then, the regression model would collapse to zero, and this can also happen in extreme cases IRL, when the columns of $Z$ are orthogonal to $Y$. Apparently, this zero model is the most robust one with respect to your objective - the error $\lVert Z\theta_{adv}- Z\theta_0\rVert^2$ is trivially the smallest when $Z\theta_{adv}=Z\theta_0=0$. This trivial solution probably arises because you determine $\theta_0$ and $\theta_{adv}$ in dependence of $Z$ and then you optimize $\lVert Z\theta_{adv}- Z\theta_0\rVert^2$ over $Z$. There is no requirement in your objective that $Y$ has to be approximated well by $Z$. To make this work, you probably have to add a constraint to your objective like $\lVert Z\theta_0 - Y\rVert\leq \epsilon$. As far as I see it, using your equation for $Z\theta_0$, the constraint would be equivalent to $\lVert Y \rVert^2 -\epsilon \leq \lVert U_{\cdot:d}^\top Y\rVert^2$. $\lVert U_{\cdot:d}^\top Y\rVert^2$ is maximized when one vector $U_{\cdot i}=Y$. The other column vectors of $U$ have then to be orthogonal and we get $\lVert U_{\cdot:d}^\top Y\rVert = \lVert Y\rVert$. Back to the original objective, the singular value $\sigma_i^2=s_0$ should then be as large as possible and all other singular values can be zero, because $U_{\cdot j}^\top Y=0$ for $j\neq i$. So again, this contradicts your hypothesis that the most robust regression model has all $d$ singular values larger than zero.
> > > > > > > > > >
> > > > > > > > > > I might be wrong here, it's not my paper and I just briefly skizzed a proof but in the best case right now, if I made some grave mistake, we don't know the optimum to the objective and the solution to the upper bound can just be very far away from the true optimum. I know there are some optimization objectives which minimize a majorative function, but for these methods, you also need to show that the majorative converges to the original function values in some aspect. Maybe this regression example also just doesn't fit exactly to the DL case. Or you could maybe consider your research problem the other way round: if you remove all directions into which your model has low variance, then how does it decrease the sensitivity to attacks? In DL, you are not so free in your choice of the representation you can choose for the penultimate layer like in the regression example. In DL, you rather optimize over function $f$, given data $X$, such that $softmax(W^\top f(X))\approx Y$, where $W$ are the weight of the last layer.  So, you cannot just choose $Z=f(X)$ arbitrarily.
> > > > > > > > > >
> > > > > > > > > > I hope you can tighten your result in a resubmission of your work. I would like to see it accepted soon, but I think as it is, it's not correct.

---

> ### Author Response · Authors · 2021-11-26
> **To Reviewer wNJf**
>
> Dear Reviewer wNJf,
>
> Thanks for your time and valuable comments. We have responded to your latest question about Theorem 1 in detail. We would appreciate it if you could take a look at our response. As the discussion deadline is approaching, your feedback is very important to us.
>
> We hope you find the response satisfactory and increase the score accordingly. If parts of them are still unclear, please let us know and we are happy to follow up.
>
> Sincerely yours,
>
> Authors

---

### Official Review · Reviewer_gwLs · 2021-11-02

**Correctness:** 3
**Technical Novelty And Significance:** 3
**Empirical Novelty And Significance:** Not applicable
**Recommendation:** 6
**Confidence:** 3

**Main Review:**

Strengths:
1) The overall presentation of the paper is clear and easy to follow;
2) In addition to the empirical experiments, the paper also provides theoretical analysis;

Weaknesses:
1) The curve trend of standard training (ST) vs. AT in Figure 1 and Figure 2 is not quite consistent?  In Figure 1, the ST curve is almost under the AT curve; while in Figure 2, the AT curve is almost under the ST curve.

2) The paper claims that FSR can further improve the robustness of adversarial defenses, however, if we look at the results, the improvements are very marginal in most cases. So I am wondering: (1) whether this marginal improvement is worth well given we need extra computation on the SVD; (2) if we repeat the experiments many times (say 10 rounds), will these results still statistically hold?

3) How will this method work for a large-scale dataset like ImageNet? Will it cause significant computational overhead due to the SVD? And will it be efficient enough for practical settings?

4) in Figures 6, 7 in the Appendix, the trend for some of the cases seems not as obvious as Figure 4. For example, Figure 6 (e), Figure 7 (d)(f), are all close to 1. I am therefore wondering about the stability of this finding.

5) It is not clear that how it is related to the PCA; need more explanations on this part.


Minor issue: under Figure 3, there is a sentence “We could compare r(D, D, u_j)…” should be “r(D_{adv}, D, u_j)”.







#-------------------------------------------------------------------------------------------------------------------------------------------#

#---------------------------------------------------------------- After rebuttal --------------------------------------------------------#

#-------------------------------------------------------------------------------------------------------------------------------------------#

Thanks so much for addressing my previous concerns.

Based on the feedback and the available reviews and discussions, I will keep my current score because 1) this method might not be practical to handle large-scale dataset; 2) According to some of the updated experimental results when there are no adversarial attacks in the dataset (and in some types of adversarial attacks), the proposed can actually hurt the performance (compared with the AT). However, in a practical setting, one never knows whether there are adversarial attacks or not in the dataset.


**Summary Of The Paper:**

By analyzing the spectral difference between the natural and adversarial examples, this paper finds that eigenvectors with smaller eigenvalues are more non-robust and adversary trends to add more components into these directions. To eliminate the dominance of the top eigenvalues, the paper proposes Feature Spectral Regularization (FSR), which adds more penalties to the largest eigenvalues while relatively increasing the smaller ones. Several experimental results demonstrate that FSR can further improve the robustness when combined with other adversarial defenses.

**Summary Of The Review:**

The overall presentation of this paper is clear and the authors have conducted experiments on different datasets, adversarial attacks, and different adversarial defenses. However, I still have several concerns as I list above, so I will give it a “6: marginally above the acceptance threshold”.

---

> ### Author Response · Authors · 2021-11-19
> **Response to Reviewer gwLs**
>
> Thank you for the supportive review. We respond to each of your questions and concerns one by one in what follows. If parts of them are still unclear, please let us know and we are happy to follow up. A revision of our paper has also been uploaded.
>
> **Question 1: The curve trend of standard training vs. adversarial training in Figure 1 and Figure 2 is not quite consistent.**
> The difference comes from that the curve in Figure 1 is regarded as a sketch map. In Figure 1, to clearly demonstrate the difference of intrinsic dimension (ID), we mark the ID near the horizontal axis, and the eigenvalues whose indices are larger than ID are regarded as 0. However, all the eigenvalues and ID are strictly calculated from the real dataset CIFAR-10 in Figure 2. In Figure 2, after normalizing the eigenvalues by scaling the maximal eigenvalue to be 1, the AT curve is under the ST curve because the latter quickly drops to zero. Thanks for pointing out this inconsistency between figures, we will modify the curve in Figure 1 to be consistent with Figure 2.
>
> **Question 2: The improvements are very marginal in most cases. (1) whether this marginal improvement is worth well given we need extra computation on the SVD; (2) if we repeat the experiments, will these results still statistically hold?**
> (1) Extra computation on the SVD. We test the total training time on CIFAR-10 using ResNet-18. Adding FSR or not actually consume nearly the same training time, i.e., approximately 19.5 hours under our experimental conditions. Since we only need to calculate the singular value, it does not increase distinctly computational cost. We have provided the analysis about computational complexity in Section 4.1, revealing that computation of SVD based on mini-batch is negligible compared to adversarial training.
> (2) Standard deviations (STDs). We rerun all the experiments in Table 1-3 five times in the same settings. Test accuracy under AutoAttack is listed since it is the strongest attack.  The results show that we achieve a steady improvement. Detailed results are added in Appendix E.1.
>
> | Defense                      | CIFAR-10              | CIFAR-100  |SVHN|
> |---|---|---|---|
> |AT                                | 48.07 $\pm$ 0.14 |  23.93 $\pm$ 0.17 | 42.00 $\pm$ 0.20
> |AT + FSR                     |**48.91** $\pm$ 0.17 | **24.76** $\pm$ 0.14 | **44.41** $\pm$ 0.24
>
> **Question 3: How will this method work for a large-scale dataset like ImageNet?**
>  Adversarial training on ImageNet requires considerable computational cost. We have tried our best but it is much more than we could afford. The work of [Xie et al., 2019] trained ResNet-101 in approximately 38 hours with 128 Nvidia V100 GPUs. ImageNet is already a large dataset and the training time of adversarial training is much longer than standard training. Therefore, when considering adversarial robustness, ImageNet is not a preferred choice. For example, the recent works of [Bai et al., 2021, Pang et al., 2021]  also do not hold experiments on ImageNet. Low-resolution images like CIFAR-10 and CIFAR-100 are now acknowledged as standard datasets in analyzing adversarial robustness. Most of the relevant studies focus more on the performance in CIFAR, as listed in RobustBench [Croce et al., 2020].
>
> **Question 4: In Figure 6,7 in the Appendix, the trend for some of the cases seems not as obvious as Figure 4.**
> The difference comes from that we use different perturbation budgets $\epsilon$ in Figure 6, 7. However, we keep the $\epsilon=4/255$ in Figure 4. When $\epsilon$ is small, the trend of the defined *variation* is not as drastic as when $\epsilon$ is large. Figure 6 and Figure 7 are used to clarify that as the perturbation budget $\epsilon$ increases, the trend of *variation* becomes more obvious.
>
> Reference:
> [Xie et al., 2019] Feature denoising for improving adversarial robustness. CVPR 2019.
> [Pang et al., 2021] Bag of tricks for adversarial training. ICLR 2021.
> [Bai et al., 2021] Improving adversarial robustness via channel-wise activation suppressing. ICLR 2021.
> [Croce et al., 2020] Robustbench: a standardized adversarial robustness benchmark. arxiv 2020.

---

> > ### Author Response · Authors · 2021-11-24
> > **To Reviewer gwLs**
> >
> > Dear Reviewer gwLs,
> >
> > Thanks a lot for your time and valuable comments. We have tried our best to address all mentioned concerns. We hope you can take a look at our response as the discussion deadline is approaching. Please feel free to let us know any additional questions you may have.
> >
> > Sincerely yours,
> >
> > Authors

---

### Official Review · Reviewer_kop5 · 2021-11-03

**Correctness:** 3
**Technical Novelty And Significance:** 2
**Empirical Novelty And Significance:** 3
**Recommendation:** 5
**Confidence:** 4

**Main Review:**

The major contribution of this paper is the demonstrations of the connections between adversary and feature spectrum, and the development of a new regularizer applicable in adversarial training. The theoretical analysis on the other hand also supports the proposal of flattening the spectrum.

The paper is well written with adequate supports from many quantitative and qualitative results, especially those for the feature spectrums.
One of the major issues with this paper is the marginal improvement over the SOTA adversarial defense methods, which can be seen in Table 1-5.

As the SOTA adversarial defense algorithms also have the potential to flatten the spectrum of features (and many other benefits), it is unclear if the FSR will always benefit the defense, as the experiment results showed.

FSR only suppresses the largest eigenvectors, and it seems there is a gap between this operation and goal of adjusting feature spectrums. Why not suppress more eigenvalues? Some explanations would help here.

Another factor worth further discussion is the basic CNN model. It is unclear whether a strong CNN model will react differently to attack and proposed defense methods, and this could be considered in the experiments.

The dataset considered only includes low-resolution images, e.g., 32x32. It would be interesting to see if the defense is effective on high-dimensional data, i.e., 224x224 where the adversarial attack is more significant.


**Summary Of The Paper:**

The paper has presented a new method to improve the robustness of features under adversarial attacks. Authors developed a new metric for the change of features subject to attacks and key findings are eigenvectors of small eigenvalues are more inclined to change under adversarial attacks, i.e., non-robust. Authors believe the dominance of large eigenvalues and eigenvectors are the primary reasons and a way of flattening the spectrum of features would help mitigate this issue. Authors propose to suppress the largest eigenvalues during training, i.e., spectral regularization, which show positive results in adversarial defense when working together with SOTA defense models.

**Summary Of The Review:**

In brief, the paper reveals the connections between feature spectrum and non-robust features under adversarial attack and proposes an interesting regularization term for robust training. However, the new method does offer significant improvement over SOTA and was not comprehensively evaluated on high-dimensional data or using different backbone networks.

---

> ### Author Response · Authors · 2021-11-19
> **Response to Reviewer kop5**
>
> Thank you for the valuable feedback. We respond to each of your questions and concerns one by one in what follows. If parts of them are still unclear, please let us know and we are happy to follow up. A revision of our paper has also been uploaded.
>
> **Question 1: One of the major issues is the marginal improvement over the SOTA methods.**
>
> We explain the improvement of the proposed method as follows:
>  - The experiments in the paper include a wide range of attacks (PGD, C&W, and AutoAttack), defenses (AT, TRADES, and AWP), backbones (ResNet-18 and WideResNet-34-10), and datasets (CIFAR-10, CIFAR-100, and SVHN), which have consistently and steadily demonstrated the effectiveness of the proposed method.
> - We repeat the experiments in Table 1-3 five times during the rebuttal phase to calculate the mean and standard deviation. The whole results are added in Appendix E.1, which shows that FSR achieves a steady improvement. Test accuracy under AutoAttack using ResNet-18 is listed as follows. FSR could achieve a competitive improvement with recent work [Bai et al., 2021] on AutoAttack.
> | Defense                      | CIFAR-10                | CIFAR-100  |SVHN|
> |-|-|-|-|
> |AT                                | 48.07 $\pm$ 0.14 | 23.93 $\pm$ 0.17 | 42.00 $\pm$ 0.20
> |AT + FSR                     |**48.91** $\pm$ 0.17 | **24.76** $\pm$ 0.14 | **44.41** $\pm$ 0.24
>  - The proposed FSR is a simple module readily pluggable into any defense method in a few lines of codes. In this paper, we use the strong baseline like [Rice et al., 2020.], which ranks first in the original paper of AutoAttack on CIFAR-10 without extra data. In addition, FSR is motivated by the connection between spectral signatures and adversarial robustness, so it could provide some new insights to understand adversarial examples and adversarial robustness.
>
> **Question 2: FSR only suppresses the largest eigenvalue. Why not  suppress more eigenvalues?**
>
>  Thanks for pointing out this experiment, we will add it in our paper. We test the results of suppressing the largest $k$ eigenvalues. The test accuracy under AutoAttack on CIFAR-10 using ResNet-18 is reported as follows. As we properly increase the number of suppressed eigenvalues, the adversarial robustness of models could be further improved. However, if we keep increasing $k$, the robustness of the model declines. We think it is due to that $k$ has some trade-off with the weight of FSR $\beta_{\text{FSR}}$, i.e., a larger $k$ may require a smaller $\beta_{\text{FSR}}$. We will add more discussions to the main body of the final paper.
>
> | $k$| 1| 2| 4| 8| 12| 16|
> |-|-|-|-|-|-|-|
> | AA| 48.71| 48.95| 49.02| **49.17**| 48.55| 48.14|
>
> **Question 3: It is unclear whether a strong CNN model will react differently to attack and proposed defense method.**
>
> The main part of our experiments is conducted on ResNet-18, which is a commonly used network in previous studies [Rice et al., 2020; Pang et al., 2021]. In addition, we also report the results using a much larger network WideResNet-34-10 in Table 6. The results show that the adversarial robustness is improved with FSR when the backbone is WideResNet. Since adversarial training is very time consuming and its training time is almost 3-30 times longer than standard training [Shafahi et al., 2019], WideResNet-34-10 is widely acknowledged as a strong backbone in adversarial training [Rice et al., 2020; Pang et al., 2021].
>
> **Question 4: The dataset only includes low-resolution images. It would be interesting if the defense is effective on high-dimensional data, i.e, 224 $\times$ 224.**
>
>  Adversarial training on ImageNet requires considerable computational cost. We have tried our best but it is much more than we could afford. The work of [Xie et al., 2019] trained ResNet-101 in approximately 38 hours with 128 Nvidia V100 GPUs. ImageNet is already a large dataset and the training time of adversarial training is much longer than standard training. Therefore, when considering adversarial robustness, ImageNet is not a preferred choice. For example, the recent works of [Bai et al., 2021, Pang et al., 2021]  also do not conduct experiments on ImageNet. Low-resolution images like CIFAR-10 and CIFAR-100 are now acknowledged as standard datasets in analyzing adversarial robustness. Most of the relevant studies focus more on the performance in CIFAR, as listed in RobustBench [Croce et al., 2020].
>
>
> Reference:
> [Bai et al., 2021] Improving adversarial robustness via channel-wise activation suppressing. ICLR 2021.
> [Rice et al., 2020] Overfitting in adversarially robust deep learning. ICML 2020.
> [Pang et al., 2021] Bag of tricks for adversarial training. ICLR 2021.
> [Shafahi et al., 2019] Adversarial Training for Free!. NeurIPS 2019.
> [Xie et al., 2019] Feature denoising for improving adversarial robustness. CVPR 2019.
> [Croce et al., 2020] Robustbench: a standardized adversarial robustness benchmark. arxiv 2020.

---

> > ### Author Response · Authors · 2021-11-24
> > **To Reviewer kop5**
> >
> > Dear Reviewer kop5,
> >
> > Thanks a lot for your time and efforts in reviewing our paper. We have tried our best to address all mentioned concerns. We would appreciate it if you could take a look at our response. We hope you find the response satisfactory and increase the score accordingly. As the discussion deadline is approaching, your feedback is very important to us, and if there are any new questions, we can therefore reply in time.
> >
> > Sincerely yours,
> >
> > Authors

---

### Author Response · Authors · 2021-11-22
**General Response**

Dear reviewers,
Thanks a lot for your time and insightful comments. In response to the comments, we have carefully revised and enhanced the manuscript. The updates are temporarily highlighted in "blue” for your convenience to check. The major changes are as follows.
 - Modification of notations about the linear model (Section 4.2 and Appendix C)
 - Additional experiments about the standard deviations of the results in Table 1-3 (Appendix E.1)
 - Ablation study about suppressing more eigenvalues (Appendix E.2)
 - Discussion on the connection between the proposed method and Lipschitz constant (Appendix F.1)
 - Discussion on the connection between our findings and related work about adversarial examples on manifolds (Appendix F.2)

We have tried our best to address all mentioned concerns. We would appreciate it if you could take a look at our response.
Sincerely yours,
Authors

---

### Decision · Program_Chairs · 2022-01-20

**Decision:**

Reject

**Comment:**

Based on the observation that the eigenvectors with smaller eigenvalues are more non-robust (i.e., adversary adds more components along such directions), the authors propose a method called Feature Spectral Regularization (FSR) to penalize the largest eigenvalue, and as a result, the other smaller eigenvalues get increased relatively.
In this paper, in addition to FSR, theoretical analysis along with experimental results on different datasets and models were presented.
Although the proposed FSR has some merits, the major concerns from the reviewers include (1) impractical use on large-scale datasets and (2) lack of significant improvement over SOTA.
Compared with other submissions I'm handling, I have to reject this manuscript.